# Combining pangenomics and population genetics finds chromosomal re-arrangements, diversified chromosome segments, copy number variations and transposon polymorphisms in wheat and rye powdery mildew

**Alexandros G. Sotiropoulos**[1,2]*, **Marion C. Müller**[1,3], **Lukas Kunz**[1], **Johannes P. Graf**[1], **Levente Kiss**[2], **Ralph Hückelhoven**[3], **Edith Schlagenhauf**[1], **Beat Keller**[1]*, **Thomas Wicker**[1]*

**1** Department of Plant and Microbial Biology, University of Zurich, Zurich, Switzerland, **2** Centre for Crop Health & School of Science, Engineering and Digital Technologies, University of Southern Queensland, Toowoomba, Australia, **3** Chair of Phytopathology, TUM School of Life Sciences, Technical University of Munich, Freising, Germany

* alexandrosgeorgios.sotiropoulos@unisq.edu.au (AGS); bkeller@botinst.uzh.ch (BK); wicker@botinst.uzh.ch (TW)

## Abstract

Grass powdery mildews (*Blumeria* spp.) include economically important fungal crop pathogens with complex and highly repetitive genomes. To investigate the diversity and genome evolution in *Blumeria graminis*, we combined population genetic and pangenomic analyses using a worldwide sample of 399 wheat powdery mildew isolates. Additionally, we produced high-quality genome assemblies for seven isolates from wheat and one from rye powdery mildew. Using these, we compiled the first grass powdery mildew pangenome comprising 11 *Blumeria graminis* isolates. We found multiple chromosomal rearrangements between the isolates that grow on wheat, rye and/or triticale hosts. Interestingly, chr-11 showed some characteristics, comparable to accessory chromosomes such as presence/absence of large chromosomal segments and higher sequence diversity. Additionally, we identified nearly 67,000 cases of copy number variations (CNVs), which were highly enriched within effector gene families. Furthermore, we found evidence for recent and high transposable element (TE) activity, such as high numbers of TE insertion polymorphisms. Analyses of TE families showed enrichment 1 kb to 2 kb up- and downstream of effector genes, and we also found high levels of TE insertion polymorphisms between populations. Our results demonstrate that chromosomal variations, gene family expansions and contractions, and TE activity are important sources of genome diversification and diversity in grass powdery mildews. Our findings indicate that a combination of pangenomic and population genetics analyses is needed to understand drivers of evolution in plant pathogenic fungi in a comprehensive way.

**Data availability statement:** Genome assemblies, annotations and other relevant files are available at the Zenodo repository under the link: https://zenodo.org/records/18946991. Raw PacBio and Illumina sequences are available in the Short Read Archive (SRA) of the National Center for Biotechnology Information (NCBI), under the project number: PRJNA1133998. All other relevant data are within the manuscript and its Supporting information files (see S1 Appendix and S2 Appendix).

**Funding:** This project was mainly supported by grants from the University of Zurich Research Priority Program (https://www.uzh.ch/en/researchinnovation/priorities/university.html) awarded to BK. This project was also supported by the Swiss National Science Foundation (https://www.snf.ch/en) grant 310030_212428 awarded to TW. This project was also partly supported by the Centre for Crop Health of the University of Southern Queensland, through the Discovery Project DP210103869 funded by the Australian Research Council (https://www.arc.gov.au/) awarded to LeK. Work in the laboratory of RH was supported by the German Research Foundation (DFG) (https://www.dfg.de/en) —TRR 356/1 2023 grant 491090170 (A03). The funders had no role in study design, data collection and analysis, decision to publish, or preparation of the manuscript.

**Competing interests:** The authors have declared that no competing interests exist.

## Author summary

Fungi are a diverse group of organisms, with some having at times a devastating impact on important crops like wheat. Powdery mildews are common fungal pathogens of many plants including wheat, rye, and other cereals. Some species, including *Blumeria graminis* infecting wheat, have become model organisms in studies of plant-pathogen interactions. Studying their genomes can improve our understanding of the pathogen's patterns of genome evolution and virulence in order to find more efficient ways to protect crops. Using high quality genomes of a worldwide dataset of wheat and rye powdery mildews, we found multiple re-arrangements in chromosomes, as well as presence and absence of large chromosomal segments in different strains. Along with some genes, important for making these pathogens infectious on various crop lines, some genomic regions were deleted or duplicated, potentially affecting how the fungus survives and spreads. Additionally, we found that transposable elements are highly active in powdery mildews and that they are enriched near so-called effector genes, which are associated with fungal virulence. Our study shows how powdery mildew genomes have diversified during recent evolution, which has implications for future breeding of crops toward better resistance to fungal diseases.

## Introduction

Powdery mildews infecting cultivated and wild grasses include economically important plant pathogenic fungi, and those that infect crops can cause significant yield losses in agriculture [1]. These pathogens are obligate biotrophs and obtain nutrients from the infected host plant tissues via the use of various effector proteins that modify the host's metabolism. Usually, effector proteins are initially predicted *in silico* as "candidate effectors", until they are validated for a biological function as effectors.

In earlier literature, all powdery mildews infecting diverse grasses (the Poaceae) were considered as belonging to a single species, named as *Blumeria graminis* [2]. Within this taxon, a number of taxonomically informal groups, known as formae speciales (ff. spp.), were delimited to identify distinct fungi that are each specialized to one or more species or genera of the Poaceae. For example, *B. graminis* f. sp. *tritici* referred to powdery mildews infecting wheat (*Triticum* spp.); *B. graminis* f. sp. *hordei* was used to identify powdery mildew on barley (*Hordeum vulgare*); and so on [3]. A comprehensive host range testing revealed that some *Blumeria* isolates, identified as f. sp. *tritici*, infected tetraploid (durum) and hexaploid (bread) wheat varieties only; others grow better on tetraploid wheat varieties [4], and were included in the newly introduced f. sp. *dicocci*. Another newly identified group was named f. sp. *triticale* and includes isolates that infect triticale, an artificial hybrid crop between wheat and rye, as well as tetraploid and hexaploid wheat, and to a limited extent also some rye varieties [5]. A recent taxonomic study recognised some of the formae speciales as different species of *Blumeria*; and retained the binomial *B. graminis* sensu stricto (s.

str.) for powdery mildews characterised with distinct morphology and DNA barcode sequences that infect *Triticum*, *Secale*, and other species of the Poaceae [6]. Our work follows the new taxonomy of *Blumeria* sensu Liu et al. [6], and refers to ff. spp. using the designations sensu [5,7], e.g., "*B.g. tritici*", whenever needed for clarity.

During the past years, fungal genomics studies have become more feasible due to cheaper sequencing technologies. Studies on plant pathogenic fungi have highlighted differences in accessory chromosomes (e.g., in *Zymoseptoria tritici* and *Fusarium oxysporum*) [8,9], chromosomal re-arrangements (e.g., in *Cryphonectria parasitica*) [10], and Copy Number Variations (CNV)/ Single Nucleotide Polymorphism (SNP) changes (e.g., in *Puccinia graminis*) [11]. Powdery mildew genomic analyses were advanced by a number of high-quality genomes of multiple wheat and barley powdery mildew isolates that were recently determined [7,12–15]. These highlighted the diversity of effectors in *Blumeria* spp. and expanded our knowledge on grass powdery mildews in general. Population genomics has helped identify *B.g. tritici* populations resulting from hybridisation of two parental populations. These were from either the same or different formae speciales [7,16], e.g., Japanese wheat powdery mildew population, which is a hybrid between USA and CHN *B.g. tritici* populations [7]. Genomics of powdery mildew species infecting dicots, including grapevine, cucurbits, pepper, and a number of tree species [17–22], is less advanced compared to *Blumeria* spp. due, in part, to contamination and other quality issues detected in some genomes [23,24].

Transposable elements (TEs) are genetic components that can copy themselves and/or move around in the genome. There are two main classes of TEs: Class I (retrotransposons) and Class II (DNA transposons) [25]. Retrotransposons tend to increase their copy numbers more rapidly than DNA transposons due to their copy-and-paste mechanism of replications [25]. Long interspersed nuclear elements (LINEs), short interspersed nuclear elements (SINEs) and long terminal repeat (LTR) retrotransposons are widespread in the intergenic regions of many fungi [25,26]. TEs have been well studied especially in humans, *Drosophila* and wheat. However, less is known about the TEs in fungi; studies conducted so far have revealed relatively lower TE content in most other fungal genomes compared to *B. graminis* [27], and other obligate biotrophs, e.g., *Austropuccinia psidii* (the causal agent of myrtle rust) [28]. Furthermore, many fungi contain a specific pathway, i.e., repeat-induced point mutations (RIP), which disrupts TEs by introducing cytosine to thymine transitions [29,30], resulting in regions with extremely low G/C content, especially in non-genic, TE rich areas. Interestingly, *B. graminis* does not contain genes of the RIP pathway, and thus, its TEs are not affected by RIP [31]. Possibly as a consequence of this, wheat powdery mildew has a high TE content (85% of its genome), among the highest known percentage of TE content in an ascomycete fungal genome found to date [13]. These two genomic patterns make *B. graminis* a good fungal model for studying TE diversity and evolution.

The "pangenome" is defined here as the entire repertoire of genes and/or all genetic elements presenting the clade/ species studied [32–34]. The core genome contains genes present in all genomes, while additional and dispensable genes may be specific for single isolates or populations [35]. Pangenomic sequencing can help us study genome diversity on a species level by including more information about gene content, duplications and deletions of large genomic regions and other structural rearrangements. Sometimes, the addition of genes that were not found due to limitations of short-read based assemblies can be informative. Also, differences in expression can be inferred, if a given gene is present in different copy numbers between isolates (e.g., multiple copies of given gene can increase the numbers of transcripts produced). Various pangenomes of the hosts of *Blumeria* have been studied over the last few years such as that of hexaploid wheat [36] and barley [37], which provided insight into natural diversity, particularly in agriculturally important loci and genes. Similarly, there were studies on pangenomes of fungal model species [38] and model fungal plant pathogens, e.g., *Zymoseptoria tritici* [33]. With the advancement and improvements of long-read genome sequencing and assembling algorithms [39], along with more studies on the genomics of more *B. graminis* isolates [14,15,40], it has become more timely and feasible to study the pangenome of some other fungi with larger and more complex genomes, such as *B. graminis*.

Here, we present the analysis of a newly assembled grass powdery mildew pangenome based on near chromosome-scale assemblies of 11 isolates and re-sequencing data from 399 isolates derived from various powdery mildew

populations from across the globe. In this manuscript, we refer to candidate effectors (either genes or proteins) simply as "effectors" hereafter, unless we specifically mention, for example genes validated to encode avirulence (Avr) proteins which are specifically recognized by their corresponding resistance proteins. All genes that are not classified as effectors, will be referred to as "non-effector" genes. The following questions were addressed: (i) What are the levels of worldwide population genomic diversity in *B. graminis* f. sp. *tritici*, (ii) What are the structural and chromosomal differences in the genomes of different isolates, (iii) What is the extent of variation in content of effector and non-effector genes between isolates, (iv) Where are TEs located relative to genes in the *B. graminis* genome and how did TEs evolve in different isolates?

## Results

### New resources for the worldwide study of wheat powdery mildew populations

A dataset was compiled using a total of 399 isolates of *B. graminis* ff. spp. *tritici* and *dicocci* that were previously sequenced with Illumina short-read technology forming the basis of our analysis here (Table A in S2 Appendix). The dataset included the recently published Illumina short-reads of powdery mildew isolates from various parts of the world, e.g., Egypt, Russia, USA, and the Fertile Crescent [7,14,41]. The short reads produced in all these studies were used here and mapped to the genome of the reference isolate Bgt_CHE_96224 [13].

The assembled dataset included sequences of isolates from all continents, except Antarctica (Fig 1A). The majority of samples originate from the Fertile Crescent, which is the region of origin of both the wheat powdery mildew and its host [7]. The Principal Component Analysis (PCA) of the SNP data (see methods) clearly revealed distinct populations (Fig 1B). Most of the variance in the PCA can be explained by the first two principal components, loosely reflecting latitude and longitude, mostly for the Eurasian wheat powdery mildew isolates (Fig A in S1 Appendix). The powdery mildew populations in North and South America, along with the ones in East Asia seem to be separated. The three isolates from Australia group together with the USA powdery mildew isolates as described before [7]. The powdery mildew populations sampled in Europe, Egypt, the Fertile Crescent, central Russia and Kazakhstan overlap. This is in agreement with the hypothesis that there are traces of admixture between these populations, but also a gradient of geographic with genetic correlation [7].

ADMIXTURE analysis showed that K = 13 is the most probable number of ancestries according to cross-validation error (Fig B in S1 Appendix), with most populations harboring sequences from different ancestries, in some cases reflecting geographical proximity (Fig 1C). For Example, the TUR mildew population shows three main subpopulations. The one with the fewest isolates has a unique mostly unhybridized ancestry, the second one shares identity with ISR isolates and the third, largest one, appears to be a mixture between a EUR and RUS ancestry. In contrast, the EGY population shows only low admixture and two main ancestral subpopulations that are mostly unique with only low levels of admixture in the neighboring RUS, TUR, ISR and IRN mildew populations. Powdery mildew of tetraploid wheat (*B.g. dicocci*) has its exclusive ancestry at K = 13, even though it shares some genetic identity with isolates from the Fertile Crescent at K 2–12 (Fig C in S1 Appendix), which is expected due to geographical proximity. Additionally, it shares traces of ancestry with isolates of the EGY, RUS and CHN mildew populations (Fig C in S1 Appendix). For K = 13, the *B. dicocci* population looks more isolated with only traces found in the neighbor IRN powdery mildew population. Additionally, the previously described hybrid nature of the JPN powdery mildew population is visible up until K equals nine (Fig C in S1 Appendix) [7].

Furthermore, nucleotide diversity of these populations (with eight random isolates per population, Table B in S2 Appendix, in 10 kb windows) (Fig 1D) shows that the populations outside Europe, West Asia and Africa that are more isolated in the PCA (i.e., ARG, USA, JPN, CHN) have a statistically less diverse mean (e.g., ARG is statistically less diverse than CHN, and CHN is less diverse than diverse EGY). This can be explained by the isolated populations being further away from the region of origin with little genetic exchange. In this regard, they are similar to island populations, unlike the

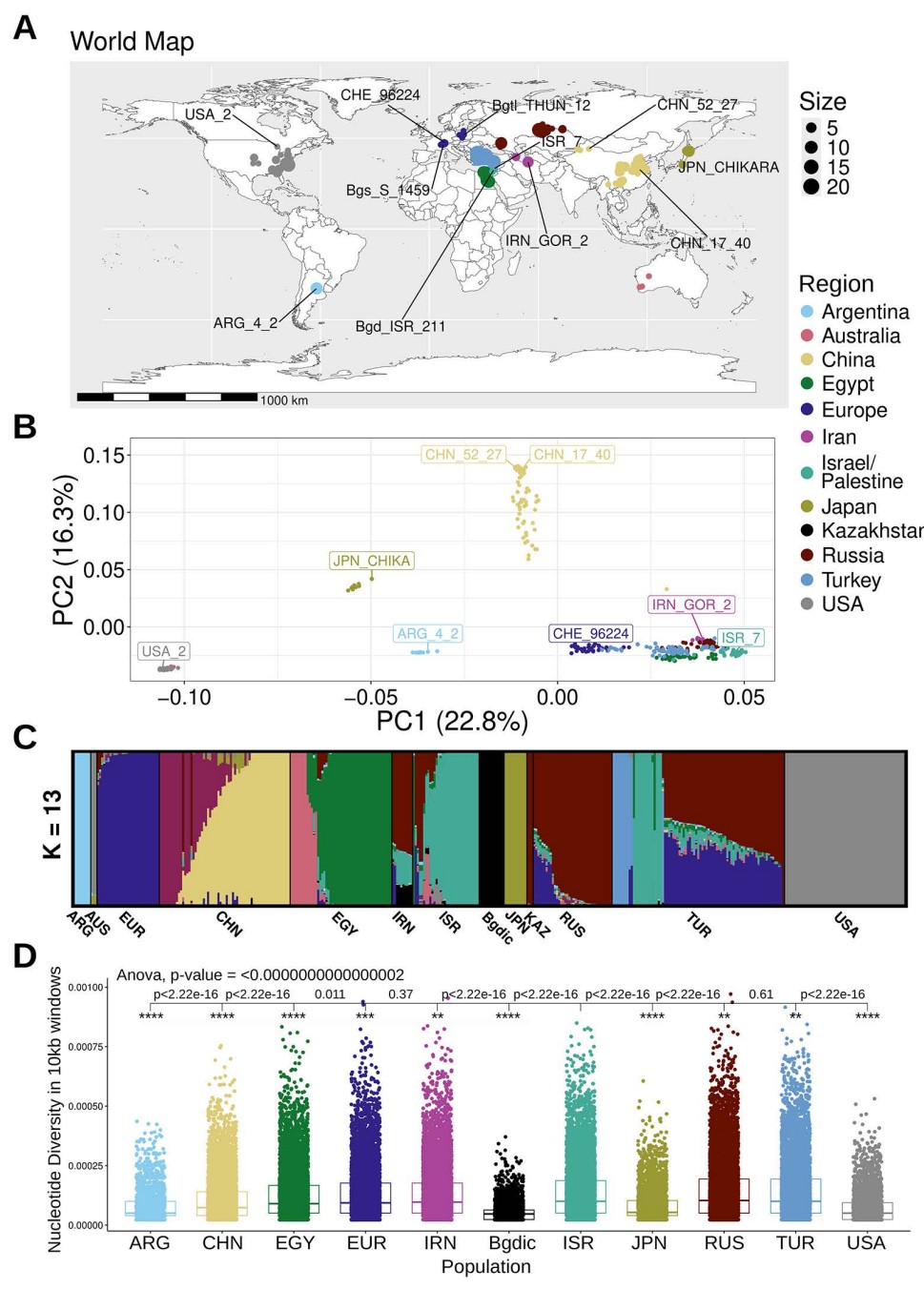

**Fig 1. Worldwide dataset of wheat powdery mildew shows varying levels of genomic diversity and clustering populations.** (A) A map with origin of the mildew isolates used in this study. Map outlines were produced with the R packages rnaturalearth and ggplot2, based on Natural Earth data (https://www.naturalearthdata.com/downloads/50m-cultural-vectors/). (B) PCA of the SNPs of 387 isolates of *B. g. tritici*. Isolates from tetraploid wheat (*B.g. dicocci*), rye (*B.g. secalis*) and rye/wheat hybrid Bgtl_THUN_12 were excluded to improve resolution of the PCA for the majority of isolates. (C) Admixture plot of the 399 isolates for K = 13. (D) Nucleotide diversity of the populations with at least eight isolates, based on eight randomly chosen isolates (see Table B in S2 Appendix) and analysing sequence windows of 10 kb. P-values represent the pairwise comparisons between neighboring pairs, while the stars show how significantly different the values are between each population and the ISR powdery mildew population.

European – West Asian – North African populations, which hold the highest diversity even as individual populations. Moreover, as previously described [7], the isolates from Australia are genetically similar to the USA powdery mildew population.

A singletons analysis, identifying SNPs unique to only one isolate within a population, supported the results of the nucleotide diversity (Table B in S2 Appendix, Fig D in S1 Appendix). A mantel test indicated that there is no significant correlation between genetic distance and geography, while using most global isolates (366 isolates, Table C in S2 Appendix, Fig E in S1 Appendix). This could be attributed to trade which could allow for recombination between populations across continents. However, when using only the wheat powdery mildew isolates in the populations around the broader region of origin (166 isolates, powdery mildew populations TUR, IRN, ISR, RUS, KAZ, Table C in S2 Appendix, Fig E in S1 Appendix), which are not separated by any major geographic barriers, such as oceans or high mountains, there is a clear positive correlation (Fig E in S1 Appendix). This result supports a previous study that used a smaller dataset [7].

### A *Blumeria graminis* pangenome representing worldwide diversity

We sequenced eight *B. graminis* isolates from around the world previously collected (Bgt_ARG_4_2, Bgt_USA_2, Bgt_JPN_CHIKA, Bgt_IRN_GOR_2, Bgt_CHN_52_27, Bgt_CHN_17_40, Bgd_ISR_211, Bgs_1459, Fig 1A and 1B, Table A in S2 Appendix) with PacBio long-read technology, resulting in chromosome-scale assemblies. For this study, we also included chromosome-scale sequences of three previously published *B. graminis* isolates (Bgt_CHE_96224, BgtI_THUN_12, Bgt_ISR_7) [13,41–43]. We used a *B. hordei* genome (isolate Bh_DH14 from England [44], GCA_900239735.1) [12] only when needed, due its fragmentation [45,46]. While eight genomes belong to the forma specialis *tritici*, we also used one isolate of rye, triticale and wild emmer (tetraploid) wheat mildews in the pangenomic dataset (Table A in S2 Appendix), resulting in a pangenome of a total of 11 powdery mildew genomes with reference genome quality similar to the reference Bgt_CHE_96224. The quality of these genome assemblies was assessed using the CRAQ software, which showed high values of continuity (Table D in S2 Appendix, Methods). The assemblies had genome sizes between ~134Mb for the *B.g. secalis* isolate up to ~143Mb for the *B.g. dicocci* isolate, with the *B.g. tritici* genome sizes falling in between these numbers. (Table E in S2 Appendix). The actual genome sizes differ, largely due to highly variable repetitive sequences in the centromeric region. As in a previous study [13], centromeres were defined by the presence of the centromere-specific retrotransposon family *RII_Fuji* (Fig F in S1 Appendix), and described in more detail below.

BUSCO (Benchmarking Universal Single-Copy Orthologs) analyses showed high levels of completeness of all chromosome-scale genome assemblies, with values above 94%, 97% and 98% completeness for the Leotiomycetes, the Ascomycota and the fungi reference dataset ("Fungi Odb10"), respectively (Figs G and H in S1 Appendix). The different genomes contain between 7,563 and 9,454 genes (Fig I.A in S1 Appendix), with the effector genes constituting ~10% of the genes annotated, while the protein size distribution of the encoded effectors was consistent among isolates (Fig I.B in S1 Appendix).

All newly sequenced isolates have 11 chromosomes as was found in the Bgt_CHE_96224 reference isolate (Figs 2, J in S1 Appendix and Table E in S2 Appendix) [13].

Three of the newly sequenced isolates in this study have few small unassembled contigs (including Bgt_JPN_CHIKA, Bgt_CHN_17_40, and Bgt_CHN_52_27, see Table E in S2 Appendix). These contained parts of the mitochondrial genome, along with un-anchored repetitive sequences. There is strong sequence collinearity between isolates, along all chromosomes, except in the highly repetitive centromeric regions (Fig 2, see also below). A comparison with barley powdery mildew (*B. hordei*) showed much weaker synteny and sequence conservation, but we still found homology along all chromosomes (Fig K in S1 Appendix).

According to a phylogenetic network created with PhyloNet using 5,982 single-copy orthogroup proteins, *B.g. secalis* forms an outgroup together with the hybrid *B.g. triticale* isolate, which has a hybrid node that indicates potential recombination. A hybrid node (which implies the potential reception of genetic material from two parents instead of one common ancestor) is also observed for the Bgt_JPN_CHIKA isolate (Fig L in S1 Appendix), as expected from previous studies [7].

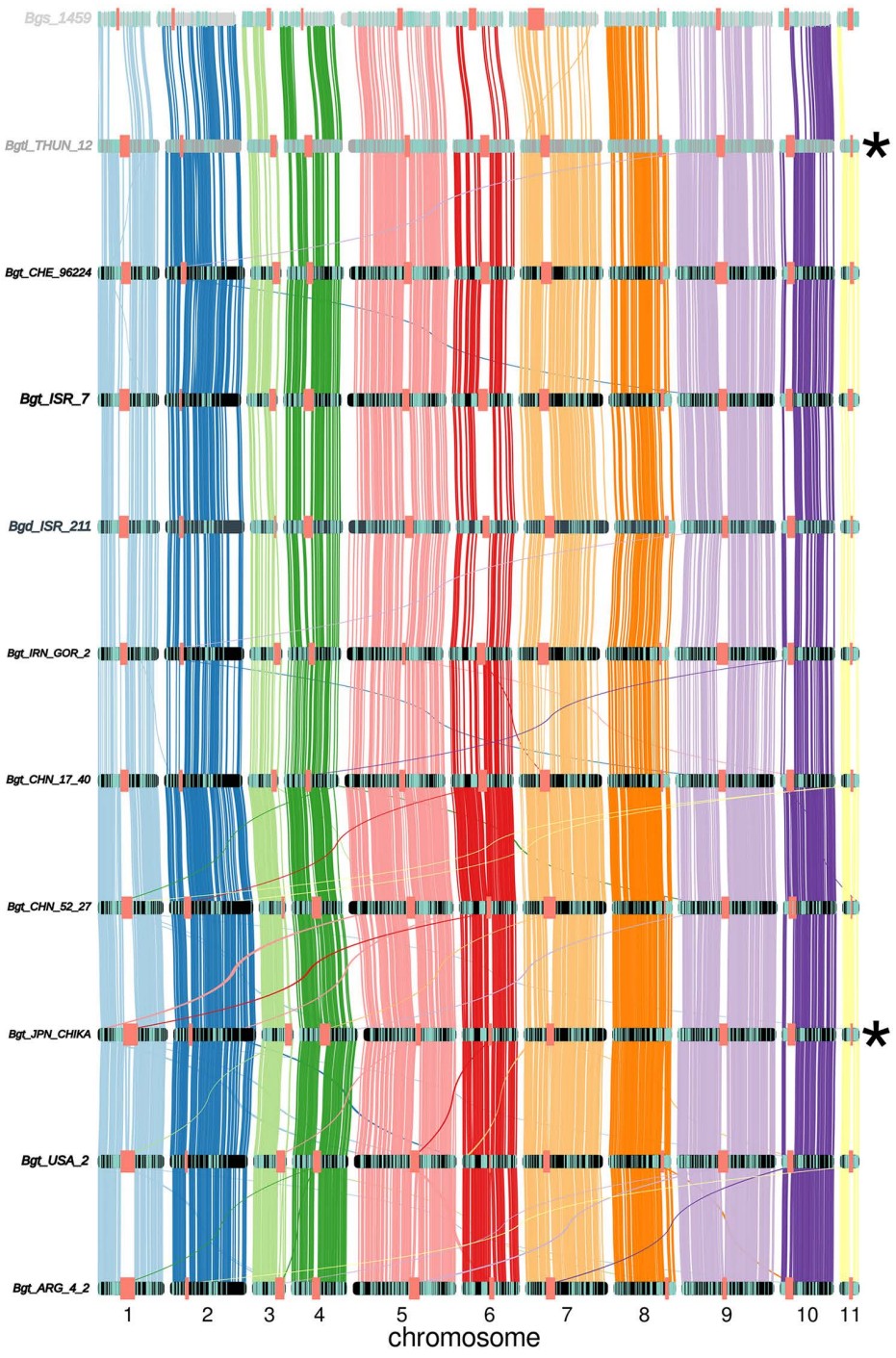

**Fig 2. Chromosomal synteny of the 11 *B. graminis* isolates in the pangenome.** The lines in colour indicate genomic regions of Minimum Alignment length of 100 kb. Each colour represents a chromosome from 1 to 11. The asterisks indicate the isolates from populations that were shown to be hybrids. The different shades of black and grey represent the different formae speciales. The red blocks indicate centromeres and the light blue ones effector genes.

We constructed a pangenome graph (Fig M in S1 Appendix) which largely reflects the chromosomal collinearity shown in Fig 2. Most chromosome arms show occasional nodes with additional sequences, while most nodes are found in centromeric regions due to high sequence diversity (Fig M in S1 Appendix, see also below). Additionally, multiple sequence nodes are found in terminal regions of chromosomes, e.g., chromosomes 5, 7 and 11, (Fig M in S1 Appendix, Table F in S2 Appendix), indicating the presence of unique chromosome segments in some isolates.

Average Nucleotide Identity (ANI) analysis in all chromosomes showed varying levels of homology: all the 11 *B.g. tritici* isolates sequenced to chromosome scale share individually more than 99.45% homology with Bgt_CHE_96224. The *B.g. dicocci* isolate shares ~99.34%, the *B.g. secalis* ~98.75%, and the *B.g. triticale* ~99.44% average homology with the original reference *B.g. tritici* Bgt_CHE_96224 assembly (Table G in S2 Appendix). This result reflects the evolution and phylogenetic distances of the respective isolates as expected.

The sequence of the mitochondrial genome for the isolate Bgt_CHN_17_40 was assembled in one contig of ~102 kb (Fig N.A in S1 Appendix). The mitochondrial genome sequence shows high collinearity with the previously published *B. hordei* DH14 and Bgt_CHE_96224 mitochondrial sequences (Fig N.B in S1 Appendix, Table G in S2 Appendix) [12,47]. We annotated 168 open reading fames (ORFs) in the Bgt_CHN_17_40 mitochondrion genome, similar to the numbers we obtained when we annotated the previously published mitochondrial sequences from *B. hordei* (163 ORFs) and Bgt_CHE_96224 (156 ORFs) (Table H in S2 Appendix) [12,47].

We also reviewed the mating type of these genomes, by using blast search of the published mating type loci on the genomes (Table I in S2 Appendix), using Bgt_CHE_96224 (MAT 1-2-1: *Bgt-3306* and SLA-2: *Bgt-2805*) [48], and the MAT 1-1-1 locus with the Bgt_CHN_52_27 gene *BgtCHN52_27_00382*. All the mating type loci were nearly identical between isolate and formae speciales with no more than one SNP difference. Both mating types were found in these 11 genomes, four isolates with MAT 1-1-1, and seven with MAT 1-2-1. The mating type locus is on chromosome 1. Sequences flanking the mating type loci are highly collinear between isolates of the same mating type, with the exception of occasional TE polymorphisms. In contrast, a region of ~200 kb surrounding the mating type genes do not align between isolates of different mating types.

## Chromosomal and structural re-arrangements within *B.g. tritici*

An interchromosomal translocation was found between the isolate Bgt_JPN_CHIKA (belonging to the Japanese wheat powdery mildew population) and the isolates Bgt_USA_2 and Bgt_CHN_52_27, which belong to the respective parental populations of the hybrid Japanese wheat powdery mildew population (Fig 3A) [7]. This interchromosomal translocation affected a ~640 kb segment containing 55 genes that was moved from chr-05 of Bgt_CHN_52_27 to chr-01 of Bgt_JPN_CHIKA, starting at ~3.5Mb of chromosome 5. This segment corresponds to the left end of chr-01 in the Bgt_JPN_CHIKA genome. This translocated region is also in chr-05 in the Bgt_USA_2 genome (Fig 3A). Thus, we assume that it is a recent translocation in the Bgt_JPN_CHIKA genome (Fig O in S1 Appendix). Furthermore, another translocation was found between chr-05 of Bgt_CHN_52_27 and chr-02 of Bgt_JPN_CHIKA, which amounts to approximately 242 kb and includes 27 genes (Fig 3A).

There are also re-arrangements between isolates coming from the same powdery mildew population (i.e., the Chinese powdery mildew isolates Bgt_CHN_52_27 and Bgt_CHN_17_40) (Fig 3B). In chr-11 of Bgt_CHN_17_40, there are two segments with possible re-arrangements between chr-01 and chr-02 of isolate Bgt_CHN_52_27. Despite high contiguity scores from CRAQ analysis for the new assemblies (Table D in S2 Appendix), we examined the breakpoints of translocations for the presence of sequence gaps for all the studied re-arrangements. We only found one gap next to a potential translocation breakpoint in chr-02 of Bgt_CHN_52_27 (position: 10,455,965 bp, Fig 3B), which is why a possible miss-assembly cannot be excluded here.

Such small structural re-arrangements between isolates are expected in a fungus with an annual sexual recombination cycle. However, more pronounced re-arrangements were found between different f. sp. isolates, as will be described hereafter.

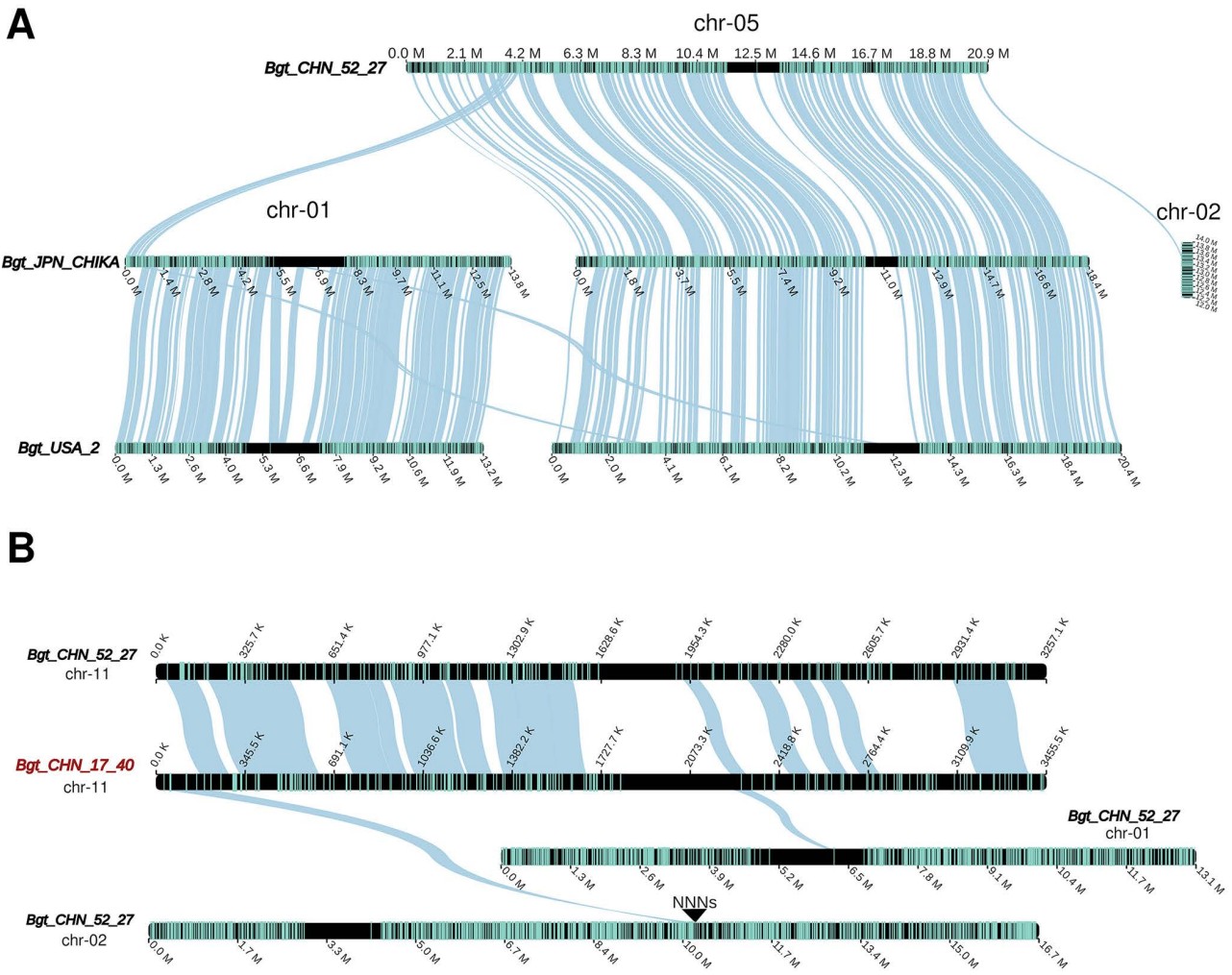

**Fig 3. Examples of chromosome collinearity and re-arrangements within *B. g. tritici*.** (A) Example of synteny and translocations in the genomes of three isolates (two coming from the parental populations, USA and CHN, and the middle genome coming from an isolate of the hybrid population of JPN powdery mildew. (B) Example of synteny in the genomes of two isolates from the same population (CHN powdery mildew). The petrol lines on the chromosomes indicate genes. The triangle and the "NNNs" show the position of a sequence gap.

## Chromosome 11 shows high sequence diversity and the presence of potentially dispensable chromosome segments

In *B.g. tritici*, *B.g. dicocci* and *B.g. triticale* isolates, chromosomes 1–10 show higher sequence conservation (ANI median ~99.38%, Fig P in S1 Appendix), while conservation is lower between *B.g. tritici* genomes and Bgs_1459 (ANI median ~98.65%, Figs P and Q in S1 Appendix and Table G in S2 Appendix), reflecting their phylogenetic distance (Fig L in S1 Appendix). However, striking differences were found in chr-11: *B.g. tritici*, *B.g. dicocci* and *B.g. triticale* isolates show an ANI median value of ~ 99.16%, while the ANI median value was lower (median ~96.98%) in comparison with the Bgs_1459 chr-11(Figs 4, P in S1 Appendix, examples in Fig Q in S1 Appendix). Additionally, sequences on the right arm of chr-11 are less conserved than on the left arm (Fig 4A), both between *B.g. tritici* isolates and between *B.g. tritici* and *B.g. secalis* (Fig 4B and 4C).

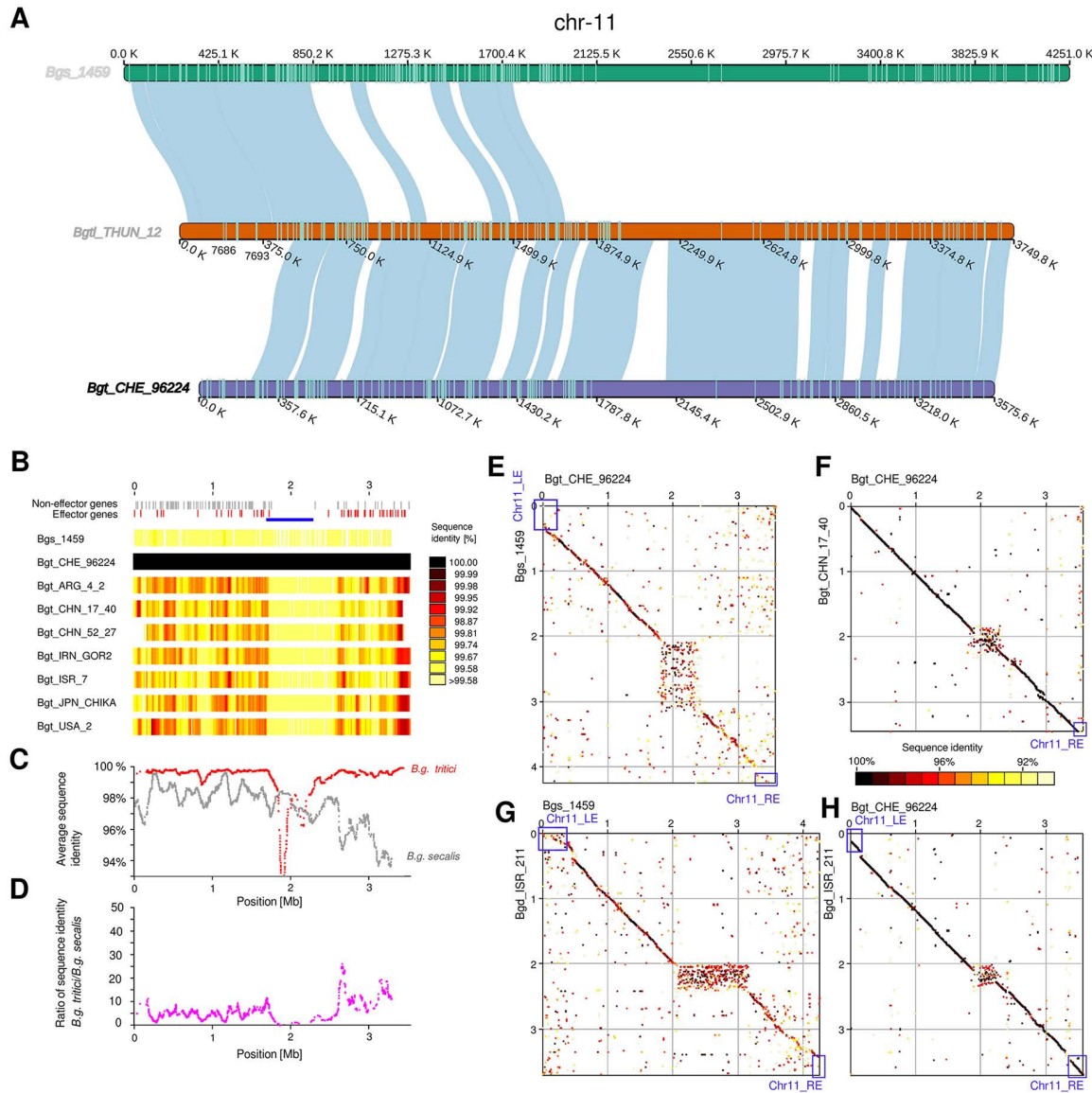

**Fig 4. Analysis of structural variation and sequence diversity on *B. graminis* chr-11.** (A) Synteny in chr-11 for the hybrid *B. g. triticale* genome and its two-parent f. sp. genomes. The petrol lines on the chromosomes indicate genes. The colour of the names of the isolates corresponds to the forma specialis (black = *tritici*, darker grey = *triticale*, lighter grey = *secalis*). (B) Sequence conservation along chromosome 11. The heat map shows comparisons of *B. graminis* isolates with the reference isolate Bgt_CHE_96224 in 50 kb windows. Bgtl_THUN_12 and Bgd_ISR_211 were excluded from this analysis. (C) The red line shows the average sequence conservation for all 50 kb windows among *B.g. tritici* isolates, while the gray line shows sequence conservation between Bgt_CHE_96224 and Bgs_1459. Note that sequence conservation is generally lower in the right chromosome arm. (D) Ratio of sequence conservation between *B.g. tritici* isolates and Bgs_1459. (E-H) Dot plot comparisons of chr-11 from *B. graminis* isolates. *B.g. secalis* and *B.g. dicocci* both have a segment at the left end (Chr11_LE, blue box) which is absent in *B.g. tritici*. Furthermore, a terminal segment of ~100 kb is absent in Bgt_CHN_17_40, suggesting it contains accessory genes.

Noteworthy are the termini of chromosome 11: as previously described, the *B.g. triticale* genome is a recombined mosaic of the *B.g. tritici* genome (~87,5%) and the *B.g. secalis* genome (~12.5%) [5,42]. In particular, in a previous study [42], the beginning of the left arm of chromosome 11 was found absent in *B.g. tritici* isolate Bgt_CHE_96224. Here, we

studied this in more detail, and indeed found that the terminal ~400 kb in Bgtl_THUN_12 originated from *B.g. secalis*, while the rest of the chromosome shows stronger synteny with *B.g. tritici*.

We found that the *B.g. secalis* and *B.g. dicocci* isolates both have this segment at the left end (Fig 4E, chr11_LE), while it is absent in all *B.g. tritici* isolates. Conversely, an ~100 kb segment at the right end of chr-11 is absent in *B.g. secalis* when compared to *B.g. tritici* (Fig 4E, Chr11_RE). Interestingly, this segment is present in the *B.g. dicocci* isolate (Fig 4G-4H), suggesting *B.g. dicocci* might represent the ancestral state as it has both the segment on at the left and right end of chr-11. Furthermore, a terminal segment of ~100 kb on the right end of the chromosome is absent in Bgt_CHN_17_40 (Fig 4F, Chr11_RE). This segment is also absent in Bgt_CHN_52_27 (Fig R.A in S1 Appendix). Additionally, isolate Bgt_CHN_52_27 is missing a ~180 kb segment on the left side of chr-11 (Fig R.A in S1 Appendix), while Bgt_ARG_4_2 has an extra segment of about ~150 kb on the left side compared to Bgt_CHE_96224 (Fig R.B in S1 Appendix). We, therefore, propose that the termini of chr-11 contain genes, which are dispensable for general pathogenicity on Triticeae host, but may be involved in host specificity.

We also want to emphasize that sequence alignments are generally poor in centromeric regions in all chromosomes. Examples are shown in Fig 4E through 4H, where centromeres are the highly variable regions. This is likely due to high number of polymorphic insertions of the centromere-specific *RII_Fuji* retrotransposons (Figs 4E-4H, and F in S1 Appendix). Our previous study showed that centromeric regions are defined by absence of genes and high enrichment of *RII_Fuji* retrotransposons [13]. The poor alignments of centromeres suggest high *RII_Fuji* activity and thus high sequence turnover. This also explains the abundance of nodes in the pangenome graph in centromeric regions (Fig M in S1 Appendix, examples in Fig 5A).

## The right arm of chromosome 11 is enriched in effectors and shows high levels of sequence variation

The right arm of chromosome 11 stands out for its diversity with respect to gene-density and content of effector genes. While all chromosomes, including the left arm of chromosome 11 contain between 6.4 and 18% effector genes, the right arm of chromosome 11 has 31 effectors out of a total of 44 genes (70.5%). Additionally, the pangenome graph shows a high number of nodes and edges compared to other chromosomes (Fig 5). Moreover, overall gene density is roughly half that of other chromosomes (Fig 5). Note that for these calculations, we excluded previously defined centromeric regions [13], in order not to distort the numbers for shorter chromosomes.

Using Illumina sequence read coverage for the 399 mildew isolates, we identified gene presence/absence polymorphisms (see methods). Generally, presence/absence polymorphism are found along all chromosomes, with effector genes having more than twice the rate of deletions and duplications as non-effector genes (Figs 5B, and S in S1 Appendix). Still, the right arm of chromosome 11 has more than double the rate of presence/absence polymorphism than chromosomes overall (Fig S in S1 Appendix). Interestingly, genes on the right arm of chromosome 11 also have the highest numbers of protein variants across the 399 isolates (Fig 5C). For this analysis, we predicted protein sequences for all genes across the 399 isolates from SNP data derived from mapping Illumina reads to the Bgt_CHE_96224 reference genome. It is therefore possible that some isolates contain additional genes and gene variants that are not present in the reference genome.

We produced a genealogic tree of all the predicted candidate effector proteins found on the right side of the chromosome arm of the Bgt_CHE_96224, Bgd_ISR_211 and Bgs_1459 isolates (Fig T in S1 Appendix). We also included the effectors found in the exclusive non-*B. g. tritici* left chromosome arm segment. Most of these genes belong to the previously described large effector family E003 [13]. Furthermore, we included effector proteins from chr-06 that had homology with candidate effector *Bgs1459–09640*, a gene that is found on the left end segment in *B.g. secalis*. This group of effector genes has apparently seen an expansion specifically in *B.g. dicocci* (Fig T in S1 Appendix). Furthermore, a gene *Bgs_09664* from the left end of chr-11 of *B.g. secalis*, groups with homologs from the right end of chr-11 in *B.g. tritici* (Fig T in S1 Appendix), indicating that the ancestral chr-11 had E003-family effectors on both ends, which underwent differential

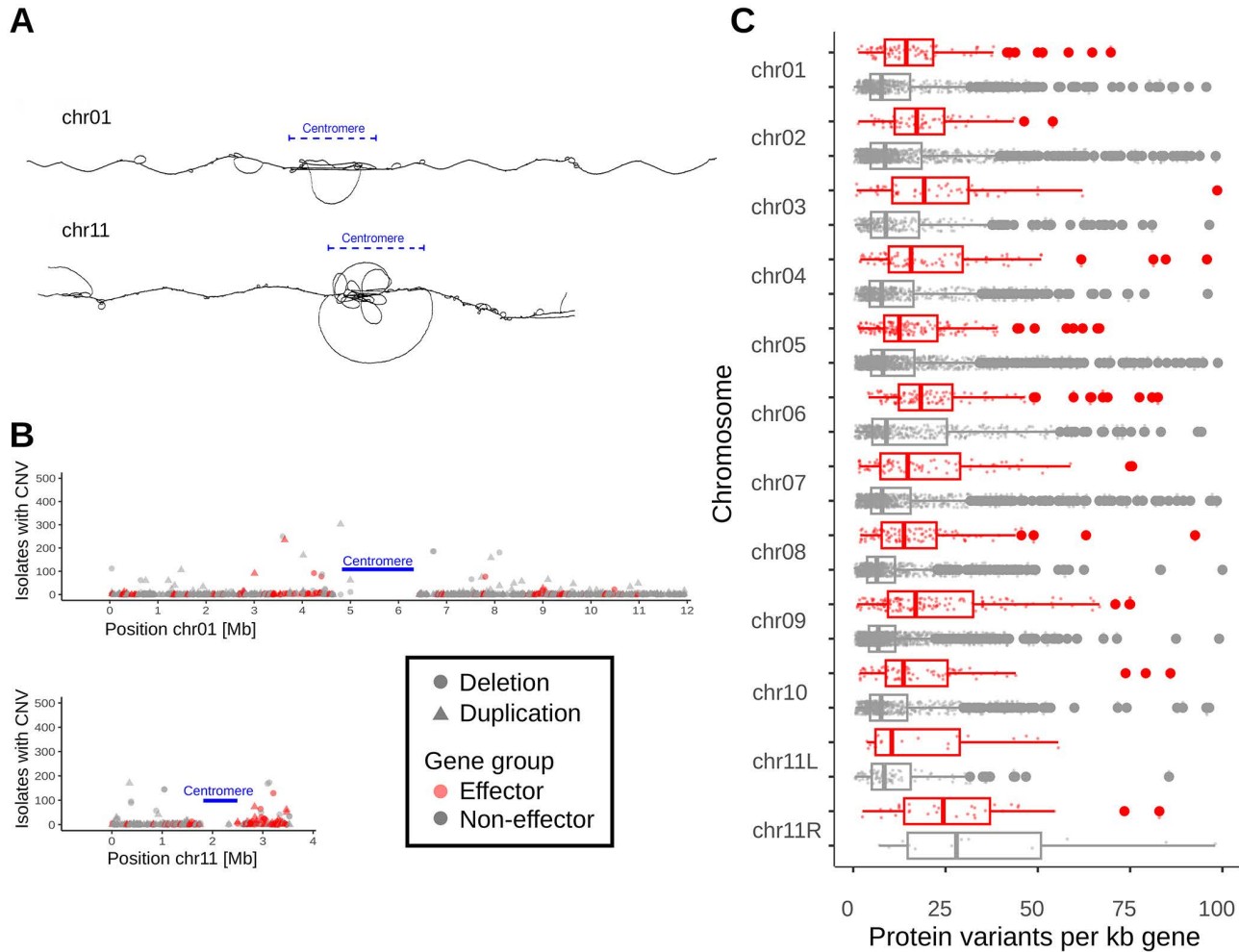

**Fig 5. Analysis of sequence variation on chromosome 11.** (A) Comparison of pangenome graphs for chromosomes 1 and 11. Approximate location of centromeres are indicated. Note that the right arm of chromosome 11 contains a higher number of nodes and edges than the left arm and than chromosome 1 overall. Additionally, both ends of chromosome 11 contain multiple edges reflecting the presence/absence of large segments between isolates (see also Fig 4). (B) Copy number variation (CNV) along chromosomes 1 and 11. The x-axis indicates the position while the y-axis shows the number of *B. graminis* isolates that show copy number variation in a given gene. Circles indicate multiple copies while triangles indicate deletions. Genes were separated into effectors (red) and non-effector genes (gray). (C) Number of identified protein variants in 399 *B. graminis* isolates per gene, normalized for gene length. Data sets for effector and non-effector genes are shown separately, so are data for the left and right arms of chromosome 11. Boxes indicate the inter-quartile range (IQR) with the central line indicating the median and whiskers indicating the minimum and maximum without outliers, respectively. Outliers were defined as minimum − 1.5*IQR and maximum + 1.5*IQR, respectively. Whiskers indicate the standard error.

loss in the different formae speciales (with 28 E003-family effectors in chr-11 for Bgt_CHE_96224, 20 for Bgs_1459 and 33 for Bgd_ISR_211).

Finally, three additional smaller segments which originate from *B.g. secalis* in *B.g. triticale* were found in the ~1.6 Mb at the left end of chr-11 (Fig 4A). This segment contains a candidate effector gene that is only found in *B.g. secalis* and *B.g. triticale* (*Bgs1459–09656* and *BgTH12–07698* respectively), but is absent in *B.g. tritici* (Note A and Fig U in S1 Appendix).

We hypothesize that presence/absence of chromosome segments, differential gene losses and family expansions of candidate effectors in chr-11 may be among the factors that contribute to host specificity.

## A variety of gene duplications and deletions in the *B. graminis* pangenome

The structural variations in chr-11 inspired us to further study presence and absence of genes in the pangenome. Here, we used short read sequence coverage data mapped on to the Bgt_CHE_96224 reference genome to identify possible gene copy number variants (CNVs). If coverage with sequence reads was particularly high for a gene in a given isolate (i.e., roughly multiples of the average sequence coverage), we took this as evidence for multiple copies of that gene. Using a normalized coverage cutoff of 0.1 < coverage < 1.7 (guided by gene coverage, see Fig V in S1 Appendix and Methods), we identified 66,856 individual CNVs for the various isolates (Figs 6A, W in S1 Appendix). We observed that the number of duplicated genes was statistically significantly higher in a few populations (e.g., EGY population) and between formae speciales (Fig 6A). For example, *B.g. secalis*, has approximately 2.5 times more gene duplications than the *B.g. tritici* isolates, while *B.g. dicocci* has approximately 1.5 times more. These results could be attributed to the larger genomic differences between the different formae speciales, but also to using Bgt_CHE_96224 as the reference genome, which creates a reference bias and possible underestimation of CNV frequencies. Furthermore, even within *B.g. tritici* isolates, we found that populations from the greater Fertile Crescent area were among the most diverse (Fig 6A).

We identified 3,795 genes with indications for CNV in at least one isolate. Fixation rates of presence/absence polymorphism in individual populations were relatively low, ranging from TUR, where only two duplications and two deletions were fixed in the whole population to *B.g. secalis*, where 211 deletions and 142 duplications were fixed in all isolates.

We created a PCA from a matrix of sequence coverage of each effector gene per isolate for all isolates including the *B.g. secalis* ones (Fig 6B, left). The latter isolates clearly differ from the rest of the *B. graminis* isolates. If we exclude the *B.g. secalis* isolates, the PCA (Fig 6B) shows that even though most populations cluster separately (e.g., *B.g. dicocci*), they seem to overlap. Additionally, further examination revealed differences between populations and gene families (Fig X in S1 Appendix), especially in the case of effector genes (Fig X.A in S1 Appendix). Furthermore, *B. graminis* isolates other than *B.g. tritici* tend to have higher number of CNVs as expected for more distant isolates.

## Estimation of the size of core genomes for effector and non-effector genes

We estimated the size of the core genome by determining the number of bi-directional closest homologs that are conserved across *Blumeria* isolates (see methods). We estimated the size of the core genome through a sample curve in which increasing numbers of isolates were added, and numbers of genes conserved in all were counted (Fig 6C). Because isolates differ in their level of similarity to one another, all possible permutations were made for each step of the sample curve (Fig 6C). Excluding different isolates from the sampling showed that inclusion of the *B.g. secalis* isolate 1459 led to lower core genome size estimates, which was not entirely surprising, since isolate Bgs_1459 represents a different *forma specialis*. Consequently, we did not include Bgtl_THUN_12 in this analysis as it is a hybrid between *B.g. tritici* and *B.g. secalis*.

From the resulting sample curves, we extrapolated the size of the core genomes. We emphasize that the extrapolations have to be taken with caution since the sample curves do not form ideal mathematically asymptotic curves. For this analysis, we treated effector and non-effector genes separately. As shown in Fig 6C, the sample curves indicate that the core gene number of non-effector genes of *B.g. tritici* and *B.g. dicocci* converges at approximately 7,228 genes, while the inclusion of the *B.g. secalis* isolate suggests a core genome of approximately 7,079 genes, which is a difference of less than 2%. In contrast, the core effector genome of *B.g. tritici* and *B.g. dicocci* comprises ~721 genes, while inclusion of *B.g. secalis* indicates a core genome of about 615 effectors, an approximately 15% lower number (Fig 6D). In the isolates analyzed here, the *B. graminis* core genome (i.e., genes conserved in all isolates) comprises 7,086 non-effector and 624 effector genes (Fig 6E). Candidate effector genes make about 8.9% of the core genome while they account for between 14.9% and 19.3% of the accessory genome (Fig 6E). The fact that inclusion of *B.g. secalis* lead to a lower estimate of the core genome is not entirely surprising, since isolate Bgs_1459 represents a different *forma specialis*.

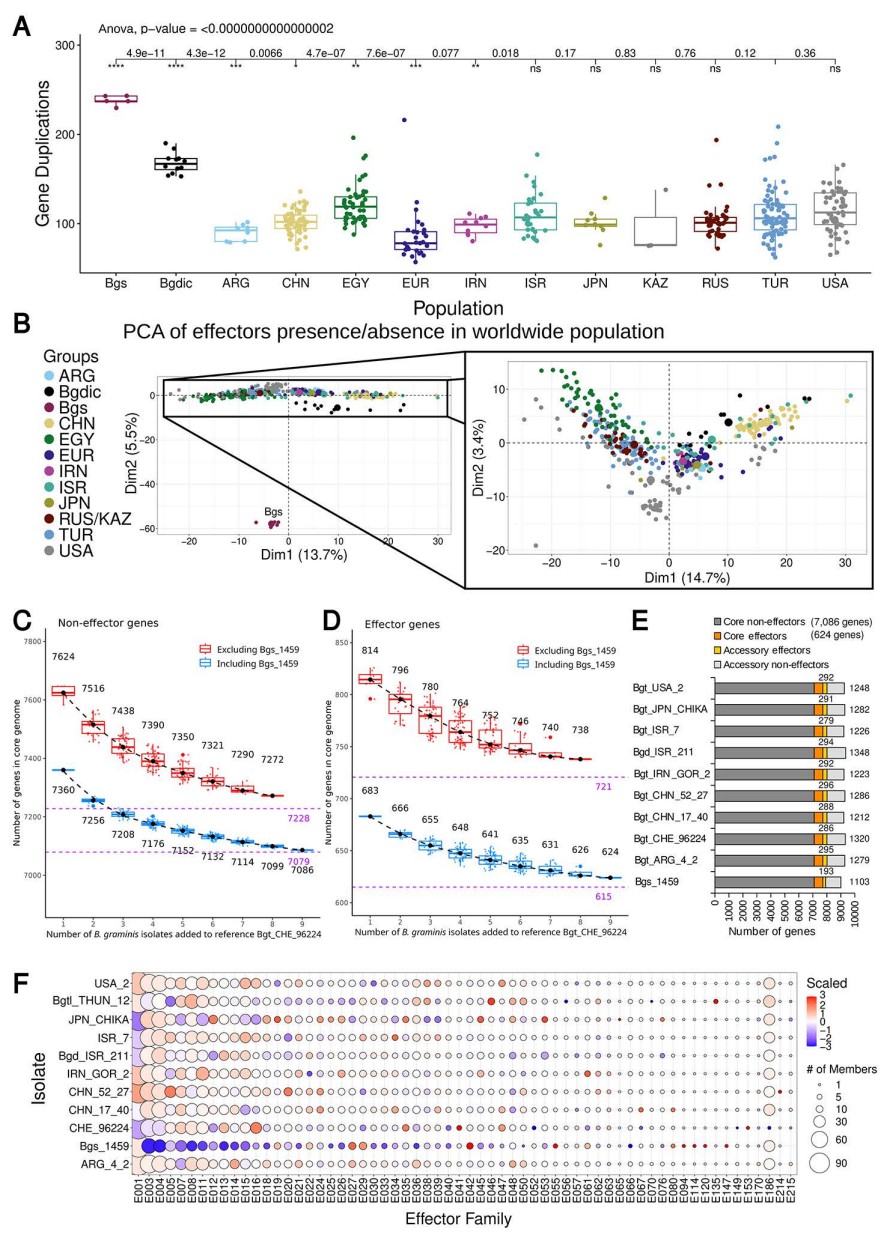

**Fig 6. Duplications/Deletions of genes using the populations and the pangenome.** (A) Number of gene duplications per population. The asterisks signify statistical significance of difference between other populations and the TUR powdery mildew population. (B) PCA of effector gene presence/absence summary per isolate, including the *B.g. secalis* isolates (left) and without them (right). (C) Estimate of the size of the core genome of non-effector genes in *B. graminis*. A sample curve was generated by adding isolates incrementally and counting the number of genes conserved between all of them. Because isolates differ in their levels of similarity, all possible permutations were performed. Two sample curves were generated, one where the *B.g. secalis* isolate was excluded (red) and one where it was included (blue). The dashed purple lines indicate the extrapolated core genome size, assuming that the sample curve was asymptotic. (D) The same analysis for candidate effector genes. (E) Summary bar plot with sizes of core and accessory genomes for the isolates analyzed. Numbers of accessory effector and non-effector genes are indicated above and to the right of the bars, respectively. (F) Expansion or shrinkage of effector families in the pangenome. Number of effector genes per family in all 11 *B. graminis* isolates. The code "Bgt_" is omitted from all the *B.g. tritici* isolates here.

Because the effectors showed high variability between isolates, we then searched the individual genomes to identify potential effector gene enrichments in specific isolates (Fig 6F). The *B.g. secalis* isolate 1459, even though it seems to have gene deletions in many of the effector families, also shows expansions of a few effector gene families (e.g., E027, E029, E042, E055, E080, E094, E114, E120, E147, Fig 6F). Furthermore, Bgtl_THUN_12 has an exclusive expansion of effector gene family E135, where Bgtl_THUN_12 has four effectors, while the other 10 isolates have only one (Fig 6F). Additionally, the isolate Bgt_JPN_CHIKA has a specific expansion of family E012 (Bgt_JPN_CHIKA has 17 members, while others have between 11 and 13) (Fig 6F). Also, for some isolates of the same powdery mildew populations (CHN), there can be expansions in one isolate for one family whilst not in the other (i.e., E001 and E005).

An interesting effector family is E014, with the identified avirulence gene *AvrPm2* as a member [49]. The *AvrPm2* gene (*BgtE-5845*) was absent only in two of the pangenome assemblies (Table J in S2 Appendix), i.e., isolates Bgt_CHN_52_27 and Bgtl_THUN_12, as was also shown previously for some of these isolates [49,50]. Family E014 has more members in some isolates (e.g., Bgt_ARG_4_2), while isolate Bgs_1459, has a reduction of gene number. Despite its smaller number of genes, Bgs_1459 has some effectors in this family that are unique and do not have homologues in the other isolates (i.e., Bgs1459–02312), as shown in a phylogenetic network of this family for all the pangenome isolates (Fig Y in S1 Appendix).

## Unequal homologous recombination is a main driver of gene presence/absence polymorphisms

To study molecular mechanisms that lead to presence or absence of genes, we manually analyzed 128 insertions larger than 2 kb that were found in at least one of the 11 PacBio genomes. To ensure that orthologous loci were compared, we required that regions flanking insertion aligned over a length of at least 20 kb (see methods). Insertions that represented the same event in different isolates and those that either contain no genes, or genes that were wrongly annotated as TEs were removed, leaving a set of 14 high-confidence alignments with gene-containing insertions. Interestingly, seven of them involved predicted effector genes.

In ten cases, we could show that unequal homologous recombination (also referred to as "unequal crossing-over") was the likely molecular mechanism. In nine cases, recombination was caused by TEs, five by small interspersed nuclear elements (SINEs) and four by long terminal repeats (LTRs) of retrotransposons. In these cases, TEs from the same family and in the same orientation flanking a gene presumably served as templates for unequal homologous recombination (example in Figs 7, Z in S1 Appendix). In our dataset, TE sequences ranging from 157 to 883 bp served a recombination templates. In all cases, we could narrow down the region where the recombination occurred, based on diagnostic sequence polymorphisms (Figs 7C, Z in S1 Appendix), in some cases down to a few bp (Fig Z.B in S1 Appendix). In one case, recombination occurred in the promoter of two neighbouring genes (Fig Z.C in S1 Appendix). Three additional cases had templates of 1–15 bp flanking the deleted regions. Here, it was likely that the deletion was caused by a double-strand break that was repaired via the single-strand annealing pathway [51].

For all 14 loci, we surveyed the Illumina data for all 399 isolates. We found that fixation levels of the presence/absence of the genes are usually below 50%, except for one locus (containing genes *Bgt-51752* and *Bgt-51753*), where the absence of the genes was nearly or completely fixed in the ARG, IRN and Bgdic populations.

## Transposable element insertions correlate with host specialisation and divergence of individual powdery mildew populations

Since TEs have been shown to play an important role in powdery mildew evolution [13] and in host-pathogen interactions [52,53], we analysed the TE composition in the *B. graminis* pangenome, as well as in seven of the wheat powdery mildew populations (ARG, CHN, *B.g. dicocci*, EUR, IRN, ISR and USA populations). All 11 isolates that are part of the pangenome have similar repeat content of approximately ~55% of the genome (Table K in S2 Appendix) and similar abundance of TE superfamilies (Fig AA in S1 Appendix). On average, ~21,000 full-length and classified TEs were annotated per isolate (Fig

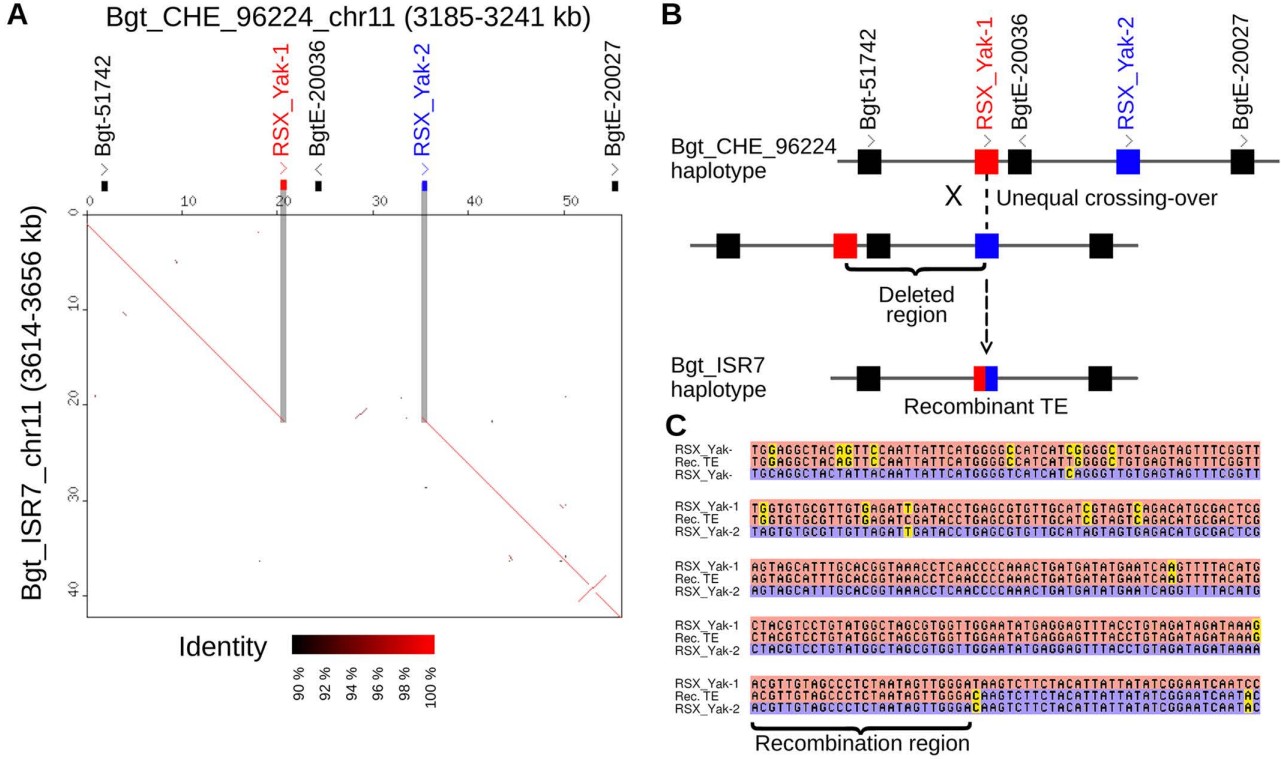

**Fig 7. Example for a gene presence/absence polymorphism caused by unequal homologous recombination.** (A) The Dot-plot alignment of the region carrying the polymorphism. Gene and TE annotation for the genome that carries additional sequence is shown above the plot with transcriptional orientation indicated by arrow heads. For easier visibility, the RSX_Yak TEs (red and blue) that served as recombination template are connected by shaded areas to the corresponding point in the plot. (B) Schematic of the molecular mechanism by which gene BgtE-20036 was deleted through unequal crossing-over. (C) Multiple alignment of the sequence that served as template. The recombinant sequence (which is found in the isolate that carries the deletion) is Rec_TE. Note that the Rec_TE sequence shares diagnostic SNPs with RSX_Yak-1 (red) in the 5' region and with the RSX_Yak-2 (blue) in the 3' region. The region where the recombination must have occurred is indicated.

8A). DNA (class 2) transposons make around 8.5% of annotated copies. SINEs (Short Interspersed Nuclear Elements) account for about 38% of the annotated TEs. LINEs (Long Interspersed Nuclear Elements) account for about 14%, while different LTR retrotransposon superfamilies account for more than 40% of annotated TEs. All TE families present in our library were found in all isolates, albeit at different abundance levels (see below). We found no evidence for TE families that are present only in certain isolates or groups of isolates.

We found an enrichment of Helitrons in the *B.g. secalis* isolate, which has 779 annotated copies, compared to the other isolates which have between 166 and 580 (Fig 8A). Moreover, the hybrid *B.g. triticale* isolate that originates from Europe has an intermediate number of Helitrons compared to its ancestors *B.g. secalis* Bgs_1459 and *B.g. tritici* Bgt_CHE_96224. We then analysed TE composition at different distances up- and downstream of genes (Fig 8B). The majority of genes are at a distance of 1,000 bp or more from the closest annotated full-length TE, with less than 3,000 TEs found closer than 500 bp to a gene on average (Fig 8B). Interestingly, TEs are closer to the effector genes, with particular enrichment of TEs around 1–2 kb distance from the effector genes.

We used the software detettore [54] to detect transposon insertion polymorphisms (TIPs) using the paired end information of Illumina read mappings of isolates to a reference genome. We ran detettore on mappings of 166 *B. graminis* isolates against the isolate Bgt_CHE_96224 (see methods and Table L in S2 Appendix). The populations closer to the region

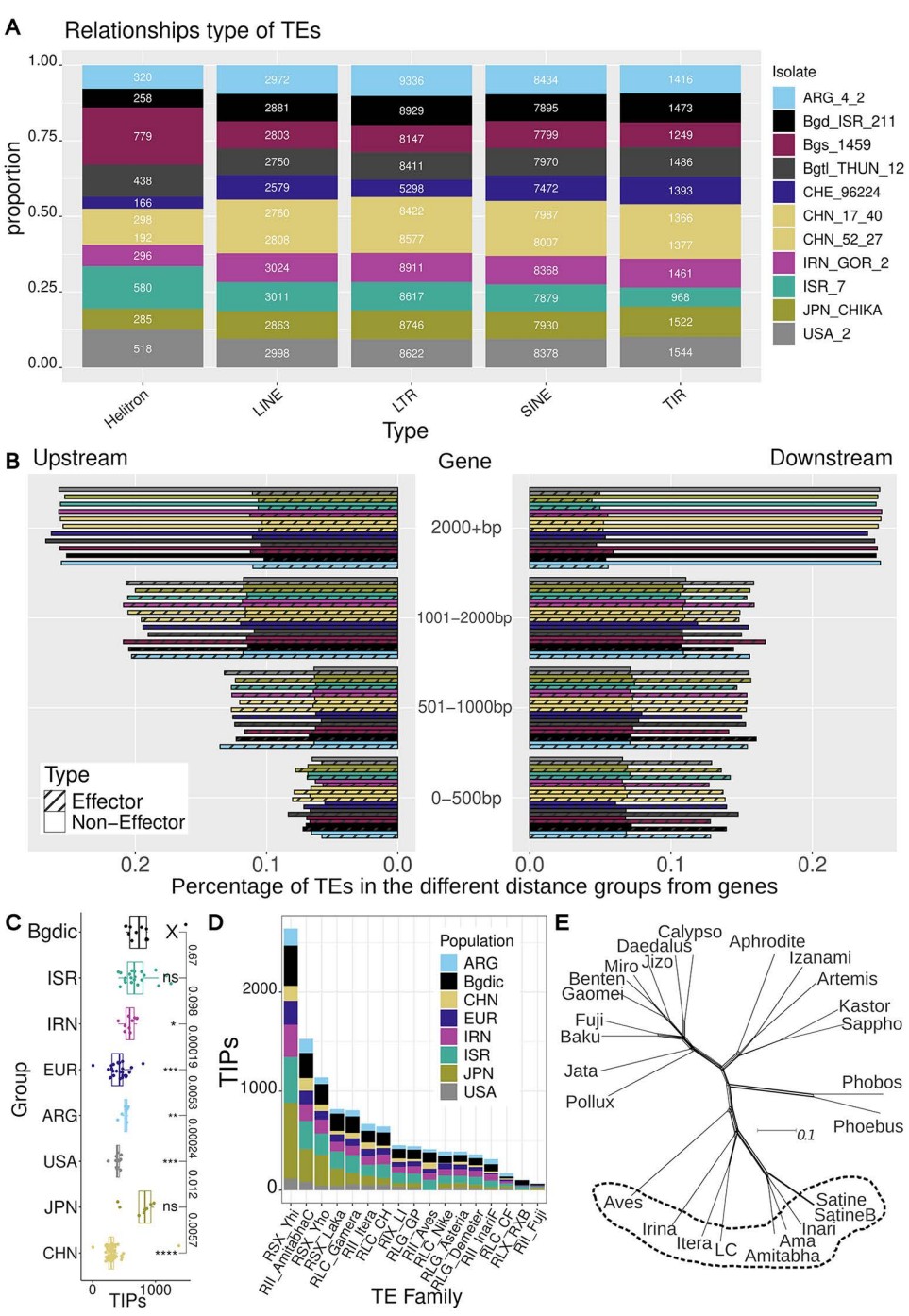

**Fig 8. TE analyses in wheat powdery mildew.** (A) Annotated copies of TE types in the different isolates of the pangenome. (B) Number of all TEs found in various distance ranges downstream and upstream of genes for all the isolates of the pangenome, see legend of Fig 8A. The bars with the dotted lines refer to distance from effector genes, while the ones without any lines refer to the non-effector genes. (C) Boxplot of the number of TIPs per isolate for populations with eight or more isolates (isolates with less than 2000 TIPs are shown, extreme outliers were excluded, because they were interpreted as technical artifacts). The "X" symbol refers to the population that the statistical comparisons have been made, were the "ns", "*", "**", "***", and "****" show significant difference with said population. (D) Analyses of TE insertions (TIPs) in the different wheat powdery mildew populations normalized for sample size for each population. Depicted are the 17 TE families that showed most insertion polymorphisms in all populations. (E) Phylogenetic network of the alignment of reverse transcriptase (RT) proteins for most of the LINE TE elements, using consensus sequences from isolate Bgt_CHE_96224. The TEs in the circle include the three LINEs with the most TIPs in the populations tested.

of origin, along with the *B.g. dicocci* population had the most TIPs (Fig 8C). The CHN powdery mildew population had the least TIPs, which can be attributed to the shorter Illumina reads, posing a bias, as shorter reads can miss TIPs due to smaller mapping space. Furthermore, the hybrid JPN population has the most TIPs. This could potentially be attributed to its hybridisation.

When looking at specific TEs, SINEs and LINEs were found to be the most mobile TEs in the various wheat mildew populations with the *RSX_Bgt_Yhi* and *RII_Bgt_Amitabha_C* families being the two with the most polymorphic insertions observed across populations (Fig 8D). The three populations in the Fertile Crescent, along with the hybrid powdery mildew population from Japan, showed the highest numbers of TIPs for most families. After LINEs and SINEs, LTR retrotransposons are the ones with most polymorphic insertions, particularly the families *RLC_Bgt_Gamera* and *RLC_Bgt_CH* (Fig 8D). The top three LINE families with most TIPs against Bgt_CHE_96224, were found across the genome (Fig AB in S1 Appendix). An interesting TE family, *RLX_Bgt_RXB* had higher numbers of TIPs in the *B.g. dicocci* population compared to all the other populations studied. This was also supported by the higher number of copies of this TE in Bgd_ISR_211 (672 copies) compared to the other genomes of the pangenome (which have fewer than 662 copies) (Table M in S2 Appendix).

Many factors such as evolutionary pressure or genetic drift could have led to this increase [55]. While it is tempting to speculate that the high number of lineage-specific TE insertions may also have driven the host-specificity of *B.g. dicocci*, more studies are needed to support this hypothesis.

Although both SINEs and LINEs generally showed high TIP numbers, only LINEs encode a conserved protein (reverse transcriptase) [25]. This allowed the study of their relative divergence time. For this, we used the predicted reverse transcriptase proteins from multiple LINE families of the Bgt_CHE_96224 isolate. The proteins were used for a multiple sequence alignment with ClustalW and construction of a phylogenetic network with PhyloNet and splitstree (Fig 8E). Some LINE proteins clustered together and were also the families with the most TE insertions (e.g., *RII_Bgt_Amitabha_C*, *RII_Bgt_Itera* and *RII_Bgt_Aves*). This indicates that the LINE cluster that includes these TEs (Fig 8E, in dotted circle) could have evolved more rapidly compared to other LINEs and could also include some of the most active LINE TEs in these genomes.

## Discussion

Recently, *B. graminis* was re-defined from a taxonomic point of view, and seven additional species were delimited within the genus *Blumeria* based on their host ranges, DNA barcode sequences, and morphology [6]. Studying the pangenomic datasets by analysing synteny and diversity has highlighted the importance of using *formae speciales* sensu Menardo et al. [5] and Sotiropoulos et al. [7] to clearly distinguish grass powdery mildews with distinct host ranges and genomic patterns within the recently defined *Blumeria graminis* species. Indeed, we found clear differences in gene content and chromosomal structures between *B.g. secalis* and the wheat powdery mildews (*B.g. tritici* and *B.g. dicocci*). However, the differences between *B.g. tritici* and *B.g. dicocci* were smaller, possibly due to their partially overlapping host ranges [16,56].

There are many facets of fungal genome architecture that play a role in the genome evolution. One of them is the presence of accessory chromosomes which could play a role in the evolution of gene clusters in filamentous some fungi such as in *Fusarium* spp. and *Alternaria* spp.[57]. Various pangenomic studies on fungal plant pathogens have highlighted their genomic diversity, like the loss of accessory chromosomes in the hemibiotrophic fungus *Zymoseptoria tritici* that can increase virulence in a specific host cultivar [58], accessory chromosomes in the hemibiotroph *Fusarium oxysporum* that indicate host specificity [59], or the high genome plasticity of the necrotrophic wheat pathogen *Pyrenophora tritici-repentis* [60,61]. However, these previous studies were done on hemibiotrophs and necrotrophs, while little is known on a pangenomic level for most obligate plant biotrophic fungi like in powdery mildews. *B. graminis* is not known to have any accessory chromosomes in a strict sense (i.e., presence/absence of entire chromosomes), which has been also supported by

the research here. However, large structural re-arrangements could contribute to reproductive isolation, and therefore, to host specialisation. We observed multiple genomic re-arrangements in this study, including the presence or absence of large chromosomal segments on chr11. Additionally, chr-11, the smallest chromosome in *B. graminis*, has the highest level of variation and some of the largest re-arrangements. This is reminiscent of characteristics of accessory chromosomes. Indeed, our study shows particular variation at the ends of chr-11 with large chromosomal segments being present or absent between formae speciales and even within *B.g. tritici*. Future studies will be needed that focus on the precise gene content of the re-arranged and additional segments to determine what role they play in adaptation and evolution of this pathogen.

Furthermore, we found numerous CNVs of genes in the *B. graminis* populations. We identified a difference in CNV frequency in genes between effectors and non-effectors on average, with a higher CNV ratio in effectors (~0.58) compared to non-effectors (~0.38), despite no specific differences in CNV frequency within individual populations, apart from the isolates that did not belong to *B.g. tritici*.

Moreover, the transposable elements landscape can have an impact on genome evolution, especially in species where most of the genome comprises of TEs like this one [13,48]. We found many TE polymorphisms in the *B. graminis* pangenome, especially between formae speciales, reminiscent of the situation in *Verticillium dahliae*, where TEs are responsible for high levels of genomic plasticity [62]. Studying TE polymorphisms showed an overall high and recent activity of TEs in *B. graminis*, which goes in line with the divergence time of these species. Previous studies indicated that *B. graminis* diverged at about 10 Ma ago from *B. hordei*, less Ma compared to each host *Triticum* spp. diverging from *Hordeum* spp. [63]. Within the *B. graminis* species, *B.g. tritici* seems to have diverged from other formae speciales (such as *B.g. secalis*) much more recently (less than 300,000 years) than its grass hosts (with rye and wheat diverging at about 4 Ma) [64]. We also found that TEs are generally closer upstream of effector genes than in non-effector genes. The association of effectors with TEs has been reported in multiple fungal pathogens [65]. We are aware that some of these results may be annotation artifacts, considering the highly repetitive nature of the genome. However, we emphasize that the reference genome Bgt_CHE_96224 had over 1,300 candidate effector genes manually curated in a previous study [13]. Possible explanations of the close association of TEs with effectors would be that effector genes simply have shorter promoters and thus allow close-by TE insertions, or that TEs themselves could act as promoters for genes, affecting expression. Another explanation would be that TEs favor open chromatin for insertion, with promoter regions of effector genes probably representing the regions with the most accessible DNA within the genome, as many effectors consistently are among the highest expressed genes [13,66]. Alternatively, relaxed selection in effector genes could allow for this to happen. Finally, TEs near effectors could potentially provide templates for unequal homologous recombination which could facilitate copy number variation and/or emergence of recombinant genes. Indeed, our manual analysis of gene presence/absence polymorphisms showed that TE sequences can serve as templates for unequal homologous recombination, if TEs of the same family and in the same orientation are located up- and downstream of a given gene. Unequal homologous recombination was previously proposed as a mechanism for gene presence/absence polymorphisms and copy number variation [67]. Interestingly, our data show that sequences of only a few 100 bp are sufficient to cause unequal crossing-over, as we could narrow down recombination break points inside these template sequences. Thus, the numerous SINEs and LTRs of retrotransposons provide abundant recombination templates. Notably, in seven of the 14 loci studied in detail, the presence/absence polymorphism involved predicted effector genes. This suggests that TEs could play an important role in the evolution of effector gene clusters. Interestingly, we only found high fixation levels for one of the presence/absence polymorphisms in the different populations. This could indicate that unequal recombination might be independently recurring multiple times. We are aware that our manual analysis allowed only the detailed study of a small number of loci. A broader, pan-genome-wide study dedicated to this topic alone will be necessary to allow more robust conclusions.

Finally, the observed association of effector genes with TEs raises the fundamental question whether the absence of RIP is a requirement for an obligate biotrophic lifestyle. Gene duplication and deletion through unequal homologous

recombination is a fundamental evolutionary mechanism to create genetic diversity. In *B. graminis* it was shown that recombination in effector gene clusters can lead to gene loss or evolution of allelic series that determine virulence and avirulence of powdery mildew strains [49,68]. Presence of RIP would both influence gene duplication and the amplification of TEs.

In general, we showed that pangenome data is suitable for identifying structural changes and large-scale chromosomal re-arrangements. Furthermore, the inclusion of short read sequencing data helped identify CNVs and presence/absence of genes and TEs in populations, even in the absence of chromosome-scale assembly for a particular isolate. These types of polymorphisms are often overlooked since most studies only focus on SNPs. Thus, we conclude that our hybrid approach of using chromosome-scale assemblies and re-sequencing data is well suited to identify different types of polymorphisms. Adding sequencing data from more isolates in the future, along with experimental evidence via chromatin capture-based methods (e.g., Hi-C) can provide an even clearer view of the genomic diversity and evolution of *B. graminis*.

## Materials and methods

### Fungal isolates collection

The whole-genome sequenced isolates were introduced into the collection by reviving the ascospores in chasmothecia [69] following the methods of [13] (Table A in S2 Appendix). The isolates that were acquired through chasmothecia were single-cell colony isolated to ensure that only one clonal individual was collected. Each isolate was propagated on ten-day-old freshly cut wheat leaves (cultivar "Kanzler") in Petri dishes. After four days, we picked with a toothpick a single colony to pass to new fresh leaves. We repeated this twice to make sure we have isolated one individual and then proceeded to propagate it, in order to get many spores for sequencing. The dates of collection of the isolates vary between 1990 and 2019 and the isolates are listed in the Table A in S2 Appendix. For each region, isolates have been collected from various fields and cultivars from *Triticum aestivum* and *T. turgidum*. A list of the isolates, the region, the coordinates of collection, the year of collection and the wheat species/subspecies that they were collected from is shown in the Table A in S2 Appendix. As outgroups for some analyses the forma specialis *B. g. secalis* was used.

### DNA preparation and whole-genome sequencing

The DNA extraction method that was used, was done with a chloroform- and CTAB based-protocol modified from Bourras and colleagues [70]. Illumina paired-end reads sequencing of 150 bp read length and an insert size of ca. 500 bp was performed to generate 1–6 Gb sequence data per isolate using NovaSeq 6000 technologies only for the isolates sequenced for long reads (see below). The rest of the isolates were sequenced before with the same extraction protocol and their accessions were found on SRA. The Illumina sequence data is accessible from the NCBI Short Read Archive (the project numbers are shown in Table A in S2 Appendix).

### Read mapping and variant calling

The raw Illumina reads were trimmed for adapter contamination and better sequencing quality. For trimming, the software Trimmomatic V0.38 [71] was used with the following settings: illuminaclip = TRuSeq3-PE.fa:2:30:10, leading = 10, trailing = 10, slidingwindow = 5:10, minlen = 50. The reference genome of *B.g. tritici* Bgt_CHE_96224 version 3.16 was used to align the reads by using the short read aligner bwa mem v0.7.17-r1188 [72] with the following settings: -M. We used samtools v1.9 [73] for sorting and removing duplicate reads. In order to mark PCR duplicates in the alignment (bam) files, we used the MarkDuplicates module of Picard tools version 2.18.25-SNAPSHOT. The average genome-wide coverage (calculated as the number of mapped reads multiplied by average mapped read length and divided by the reference genome size) ranged between x10 to x60 for all the isolates. We finally used 'samtools index' to create indices for the bam files. The mapped reads were also further checked visually for their quality and GC% content, using the software qualimap2

and the multi-bamqc function [74]. Isolates that were found to be clonal in a previous study where excluded [7] leading to 399 isolates used for most analyses.

The mapped reads were used to make variant calling files using GATK v.4.1.2.0 [75–77]. We started by using the HaplotypeCaller with the options: --java-options "-Xmx4g", -ERC GVCF, -ploidy 1. Afterwards, we used CombineGVCFs to combine the single gvcf files into one file and then, with GenotypeGVCFs we performed joint variant calls on the merged variant gvcf file using the option: --max-alternate-alleles 4. We used the GATK VariantFiltration function to hard filter SNPs with quality thresholds recommended by GATK [76,77] and used with other plant pathogenic fungi [78,79]. The following thresholds were used: QUAL < 450, QD < 20.0, MQ < 30.0, -2 > BaseQRankSum > 2, -2 > MQRankSum > 2, -2 > ReadPos-RankSum > 2, FS > 0.1. Then, we filtered for a genotyping rate (>99.9%) and removed the indels. Finally, we kept only bial-lelic SNPs using vcftools: --max-alleles 2 (version 0.1.5) [80]. We also filtered the vcf files for Minor Allele Frequency < 0.1. The number of SNPs retained was 60,258.

## Population genomics analyses

To select the isolates for long-read sequencing we performed various population genetics analyses. We used R (v4.3.3) [81], and the R packages tidyverse (v2.0.0) [82], ggplot2 (v3.5.1) [83], and stats (v4.3.3) [81] for many of the following analyses. Inkscape (v1.3.2) was used to compile many figures [84]. We first created a map with the coordinates of the samples using R packages: sf (v1.0-18) [85,86], rnaturalearth (v1.0.1) [87], ggspatial (v1.1.9) [88], and ggrepel (v0.9.3) [89]. Map outlines were based on Natural Earth data (https://www.naturalearthdata.com/downloads/50m-cultural-vectors/). Then, in order to create a PCA, we used PLINK (v2.00a3 64-bit, 17 Feb 2020) to make an eigenvec and an eigenval file. We used the file eigenval to make the PCA variance plot (Fig A in S1 Appendix), and then the eigenvec file to visualise the PCA with ggplot2 in R.

Admixture analysis was performed using the ADMIXTURE (v1.3) software [90]. With the help of the Snakemake (v9.12.0), we run ADMXITURE for K equals from 2 to 20, with 20 runs for each K and different seeds, with the parameter "--haploid = "*"". The results were visualised using the pong software [91] and the cross-validation error was visualised using ggplot2 in R.

Furthermore, we measured the nucleotide diversity of populations using vcftools (v0.1.16) with the parameter "window-pi" for 10 kb non-overlapping windows, and then visualised the results with the packages ggplot2, and ggpubr (v0.6.0) [92,93]. We used again vcftools to identify the singleton SNPs (SNPs found only in one isolate/individual within a whole population) once using all isolates and another time with only eight random isolate per populations.

Finally, we used packages vcfR (v1.14.0) [94] with the command "read.vcfR" to input the vcf file in R, then "vcfR2gen-light" to convert the object, then StAMPP (v1.6.3) with the command "stamppNeisD" to find the genetic distances between isolates and between populations [95], we inserted the geographic coordinates and transformed them into distances with "dist" from the stats package [96], we then created a two dimensional kernel density heatmap with the command "kde2d" from the package MASS (v7.3-60) [97], with the support of the packages grDevices (v4.3.3) and graphics (v4.3.3) [81] to study the genetic and geographic correlation between the isolates. The package rcartocolor (v2.1.1) was used to find colour-blindness friendly colours used in many of the plots [98].

## PacBio sequencing, gene and TE annotation

In addition to the Illumina sequencing, we long-read sequenced with the PacBio technology eight powdery mildew isolates (six *B.g. tritici*: Bgt_ARG_4_2, Bgt_USA_2, Bgt_JPN_CHIKA, Bgt_IRN_GOR_2, Bgt_CHN_52_27, Bgt_CHN_17_40, one *B.g. dicocci*: Bgd_ISR_211, and one *B.g. secalis*: Bgs_1459). There was 1x SMRT Cell 1M reads used per isolate, apart from isolate Bgt_CHN_17_40, where 8M reads with PacBio HiFi were used. The reads from the isolates were then initially assembled using SMRT Link software tools (for versions see in Table N in S2 Appendix) at the Functional Genomics Center Zurich (FGCZ) in Switzerland. The assembled contigs were then scaffolded into near chromosomes by aligning

them to the reference genome of isolate Bgt_CHE_96224, Bgt_genome_v3_16 [13]. Each contig was placed and orientated by its best blastn hit (first: e-value, second: bit-score). Contigs were merged with stretches of 200 "N" insertions (sequence gaps). Unplaced contigs are shown in Table E in S2 Appendix. We evaluated the quality of the new genome assemblies using Clipping Reveals Assembly Quality (CRAQ) (v1.10) [99]. We also used the already sequenced and assembled genomes of two *B.g. tritici* (Bgt_CHE_96224, Bgt_ISR_7) and one *B.g. triticale* (Bgtl_THUN_12) isolate for the pangenome analyses (Table O in S2 Appendix). One recently sequenced genome in the European wheat powdery mildew population (isolate CHVD_042201) was not used [100], as it was not accessible at the time this analysis was done. The assembled genome data is accessible from NCBI (the project numbers are shown in Table A in S2 Appendix). The difference between these genome was evaluated using the OrthoANIu tool (v1.2), a standalone average nucleotide identity (ANI) calculator [101]. The mitochondrial genome, the coding sequences (CDS) and individual chromosomes were also evaluated in pairs of isolates using OrthoANIu.

The gene annotations were performed using maker (v2.31.10). The annotations were generated by using the Bgt_CDS_v4_23 proteins as a template. Maker was used with the prot2genome function and the repeats and transposons were found with maker internal TE_proteins, as well as the initial annotations in PTREP and nrTREP databases [102]. On a second maker round, the annotation was created based on the *B. hordei* gene annotations of isolates DH14 and RACE1 (GCA_900239735.1 & GCA_900237765.1). From the resulting genes in the second round, only genes with no annotation in their corresponding loci in the first round (non-redundant genes) were added to the final annotation (as extra genes). In order to assess the genome and assemblies and annotations we used BUSCO (v5.4.4) [103,104] along with some further analyses below. We looked at the size of chromosomes and created a phylogenetic network using Splitstree4 (version 4.19.1, 27 Jun 2023) [105], and 100 number of replicates for bootstrapping, with the orthogroups within *B. graminis* (Fig L in S1 Appendix) via Orthofinder (v2.5.5) and the parameter "-msa" [106]. The number of genes and their type (effector, non-effector etc) was visualised, along with the protein size distribution to further check for the quality of the annotation (Fig I in S1 Appendix).

TEs were further annotated by first using the TE database for *Blumeria graminis* nrTREP (version 25) [102] and the software RepeatMasker employed by EDTA (v.2.0.1) [107]. We then manually curated newly identified TEs and performed a final annotation with the latest *Blumeria* repeats deposited in nrTREP (Table P in S2 Appendix).

The mitochondrial genome sequences were identified by using blastn of the mitochondrial contig of the *B. hordei* DH14 mitochondrial genome to the genome assembly of isolate Bgt_CHN_17_40. The mitochondrial genome was in one contig. We then, used the software MITOS2 (v2.1.9) to annotate the one mitochondrial genome with number four genetic code (Mold, Protozoan, Coelenteral [108]. The annotated output was visualised using OrganellarGenomeDRAW (v1.3.3) [109]. We also used the software Mfannot [110] (online version) with the same genetic code to re-annotate three mitochondrial genomes of interest for comparison using the previously annotated Bgt_CHE_96224 mitochondrial genome [47] and the already assembled Bh_DH14 mitochondrial genome [12].

### Synteny in the pangenome

In order to study any structural re-arrangements in the *B. graminis* pangenome, we looked at the synteny between the chromosomes and the various isolates with a Minimum Alignment Length set to 50 kb (for the individual chromosomal re-arrangement visualisation in Fig 3A and 3B) or 100 kb (for the overall chromosomal synteny in Figs 2, and 4A). For the synteny between the isolates Bgt_CHE_96224 and Bh_DH14 (Fig K in S1 Appendix), the Minimum Alignment Length was set to 10 kb, since these genomes are more dissimilar than the genomes within the *B. graminis* species. To perform these alignments, we used mummer (v4.0.0rc1) [111] between genome pairs through the command "GetTwoGenomeSyn.pl" from the software NGenomeSyn (v1.41) [112], using the appropriate minimum alignment lengths (as stated above) and the default other parameters. We then used "NGenomeSyn" command to visualise the synteny. We used the Paul Wong colour accessible palette the various isolates and populations across the rest of the study [113].

## Analysis of sequence conservation between chromosomes

To study sequence conservation between isolates along chromosomes, 1 kb segments of the reference isolate Bgt_CHE_96224 were used in blastn searches against the corresponding chromosomes of all other isolates. Blastn hits were then filtered for collinearity, requesting that a given 1 kb segment had its top blastn hit within 5% of its position in the reference chromosome. Additionally, blast hits in reverse orientation were filtered out. With this, we filtered out non-specific matches stemming from repetitive sequences. No other filtering was used, because we did not have any top blastn hits shorter than 300 bp, and even between formae speciales (e.g., between Bgt_CHE_96224 and Bgs_1459), ~94% of top blast hits were longer than 500 bp and ~79% were longer than 900 bp. Sequence conservation between Bgt_CHE_96224 and a given isolate was then calculated as a running average over 50 of the colinear 1 kb segments. Overall sequence conservation across all *B.g. tritici* isolates was then calculated as the average of all 50 kb windows in the *B.g. tritici* isolates. This was used to produce the conservation plots shown in Figs 4C and Q in S1 Appendix. Finally, the ratio of sequence conservation between *B.g. tritici* isolates, and *B.g. secalis* was calculated for all windows, resulting in the plots in Figs 4D and Q in S1 Appendix.

Chromosome-scale dot plots were produced by running blastn searches of 1 kb segments of a chromosome of one isolate against the corresponding chromosome of another. Positions of the hits in both chromosomes were recorded and used for the dot plots shown in Fig 4.

Finally, to study phylogeny of candidate effectors from chr-11 and selected homologs from other chromosomes, we created a dataset with all the candidate effector identified proteins from the right chromosomal arm of Bgs_1459, Bgd_ISR_211, and Bgt_CHE_96224, along with the effectors on the left end region that is missing in the *B.g. tritici* isolates and homologous effectors of Bgs1459–09640 found in chr-06 after using blast (v2.7.1+) [114] on these three genomes to find any homologous protein. We excluded three proteins that had a size smaller than 50 amino acids (three were excluded). We added the unclassified candidate effector protein Bgs1459–00747 as an outgroup. We aligned the dataset with ClustalW (v.2.1) [115,116], and constructed a phylogenetic tree using the Bayesian inference with MrBayes (v.3.2.7a) [117,118] with settings nst = 6 and rates = invgamma. The Markov chain Monte Carlo (mcmc) analysis was run until the probability value decreased to under 0.02 with a sampling frequency of 10 and a burn-in of 25% of samples. We visualised the tree with Figtree (v1.4.4) [119] and we used colours from Paul Tol's accessible palettes (https://sronpersonalpages.nl/~paultl/).

## Identification of specific candidate effectors

The study of some specific effectors of interest was made by first changing the annotation files gff to gtf format using gffread (v0.12.8) [120]. We then used STAR (v2.5.2a) [121] to map already published RNAseq data (namely SRR6410427, a replicate of RNAseq data of isolate Bgtl_THUN_12 on the Timbo triticale line 2 days post infection). Using IGV (v2.15.4, 12/08/2022) [122], we manually checked the expression and the annotation of specific genes BgTH12- and Bgs1459–0. After using blast (v2.7.1+) [114] to find the closest homologous proteins on the whole pangenome and on the *B. hordei* isolate DH14, we aligned them with ClustalW (v.2.1) [115,116] and visualised the alignment with Jalview (v2.11.1.4 + dfsg-3) [123]. A phylogenetic tree was constructed using the same method as described above.

## Identification of gene duplications and deletions

Identification of potential gene deletions and duplications based on Illumina sequence read coverage from the 399 isolates (including *B.g. secalis/triticale/dicocci* isolates was done as previously described [124]. The strategy here was to infer gene deletions from a lack of mapped reads for a given gene in a given isolate, and duplications from higher than expected coverage. To establish formal thresholds to determine gene copy numbers, read coverage was calculated for all individual genes across all 399 mildew isolates used in this study. For each gene, it was statistically tested whether reads coverage across the 399 isolates showed uni-modal or bi-modal distribution. The test was done with the diptest

R package (v0.77-2) and a p-value of 0.01 was used as cutoff for bimodal distributions. Positions of the modes were determined for each gene with the LaplaceDemon R package (v16.1.6). For the selection of genes with bimodal distribution, identified modes had to be below 0.2 and between 0.8 and 1.2. This was done to exclude genes that showed more continuous distributions (e.g., due to poor mapping of divergent variants or paralogs). Distributions and determination of thresholds are shown in Fig V in S1 Appendix. Using this analysis, we determined that sequence coverage below 0.1 was indicative of gene deletion, while coverage above 1.7 indicated duplication. For duplications, we did not distinguish between two or more that two copies.

### Core genome size estimates

To account for genes that might have been missed in the annotation of a given genome, the Liftoff software (v1.6.3) was used (obtained from github.com/agshumate/Liftoff/) [125]. Each of the 11 genomes was used as a target onto which annotations of the other 10 genomes were mapped. After each Liftoff mapping, new genes (i.e., genes that do not overlap with previously annotated ones) were added to the existing gff file of a respective genome. For all mappings, the "-copies" option of Liftoff was used to allow identification of duplicated genes. The cumulative Liftoff mapping of the 11 genomes was done with an in-house perl script "pan_genome_liftoff_multiple_genomes". The resulting gff files for all 11 genomes were used to produce coding sequence (CDS) fasta files using the gffread software (v0.12.7) obtained from ubuntu repositories (unbuntu.com).

To estimate the size of the core genomes (i.e., genes that were found in all isolates), the CDS of all isolates were used in bi-directional blastn searches to identify bi-directional closest homologs. This was done in a consecutive way in that the CDS of the reference genome Bgt_CHE_96224 were used in bi-directional blastn searches against CDS of all 10 other genomes. The blastn searches were run with the in-house perl script "blast_bi_directional_from_db_list". The result was a presence/absence table for all genes across all 11 genomes. This was transformed into a simple table where presence/absence was encoded as 1 or 0 with the script "pan_genome_PAP_matrix_from_bi_di_homologs". To approximate the size of the core genome, a sample curve was derived, adding increasing numbers of isolates to the reference genome and determining the number of conserved genes. Because isolates show different levels of similarity, for each sample step all possible permutations were tested. This was done with the in-house perl script "pan_genome_mk_core_genome_permutations". Sample curve data was used to calculate the convergence value (i.e., the size of the core genome) using asymptotic regression in R. Estimation of the size of the core genome was done for candidate effector and non-effector genes separately. All perl and R scripts used for this analysis are available on Github (https://github.com/wicker314/).

### Detailed analysis of loci containing gene presence/absence polymorphisms

For the analysis of mechanisms that lead to presence or absence of genes between isolates, we focused on insertions in regions where chromosomes align well between given pairs of isolates. We split the chromosomes of one isolate into non-overlapping segments of 1000 bp and used them as queries in blastn searches against the other isolate. We then searched for regions where the top blast hits were consecutive for at least 20 kb, but were interrupted by at least 2 windows that had no hits in isolate one (i.e., isolate 1 had additional sequence).

Regions where isolate 1 had additional sequence were then cross-matched with the annotation (gff files) to check whether the additional sequence carried genes. This search was run until at least 100 insertions were found (the search was stopped after 128). The candidate regions were the excised from the respective genomes (insertion region plus 20 kb flanking sequence). The excised regions were analyzed manually by dot plot (programme dotter, obtained from ubuntu.com). In multiple cases contained annotation artifacts (e.g., TE segments annotated as genes), or the same insertions were detected in multiple isolated. After these were removed, a set of 14 high-confidence insertions remained. These were searched for signatures such as overlapping sequences at the insertion breakpoint in the isolate that carries the deletion (i.e., the putative template for unequal recombination). The template sequences were annotated manually to

identify the type of repeat that served as template and aligned with ClustalW to identify diagnostic SNPs that allowed narrowing don the recombination point.

## Analyses of transposable elements for the pangenome

We analysed positions of TEs relative to genes with the TEGRiP pipeline (github: https://github.com/marieBvr/TEs_genes_relationship_pipeline) [126,127]. In order to find TE insertions which are polymorphic between isolates, we used the software detettore (v.2.0.3) [54,128] and a subset of isolates (Table L in S2 Appendix). We used ClustalW (v.2.1) [115,116] to produce the alignments of the TE consensus sequences of families of interest, and then changed the format to nexus and checked the alignment with ClustalX (v2.1) [116,129]. We used MrBayes (v.3.2.7a) with settings nst = 6 and rates = invgamma to infer trees [117,118]. The Markov chain Monte Carlo (mcmc) analysis was run until the probability value decreased to under 0.01 with a sampling frequency of 10 and a burn-in of 25% of samples. We visualised the tree using FigTree (v1.4.4) [119].

## Code availability

All R and original Perl scripts used in this study were deposited in our GitHub repository (github.com/Wicker-Lab/Blumeria_pangenome). There, we provide all R scripts that were used to produce the graphs in the main and supplementary figures, as well as the underlying data used for the plots. For Perl scripts, we provide detailed explanations of the process how they were used. Large datasets such as variant call files (vcfs) were deposited at Zenodo (https://zenodo.org/records/18946991).

## Supporting information

**S1 Appendix. Note A. Fig A. PCA variance.** Percentage of variance explained for the PCA in Fig 1. **Fig B.** Cross validation error for the ADMIXTURE analysis (for number of ancestries/"K" from 2 to 20). **Fig C.** Admixture plot of the 399 isolates for K = 1 through K = 12. **Fig D. Singleton analysis of the *B. graminis* populations.** Singletons of 400 isolates or a subset, excluding most of the clonal isolates and a few ones with very low coverage. (A) Singletons of both SNPs and INDELs for a subset of the eight isolates that were used as random ones to make up populations. Their names are listed. (B) Singleton SNPs of all 400 isolates. (C) Singleton INDELs for all 400 isolates. **Fig E. Mantel test on subsets of *B. graminis* isolates.** Mantel test using the genetic data from the SNPs and the coordinates for the geographic correlation for a worldwide dataset where geographic information is available (on the left) and for the area around the region of origin (on the right). The colors represent two-dimensional kernel density estimation of the points with red (higher density), yellow (average density), blue (lower density) and white (no density). **Fig F. Distribution of the LINE *RII_Fuji* in the genomes of three powdery mildews.** Distribution in: (A) The *B.g. secalis* Bgs_1451 genome, (B) the *B.g. triticale* Bgtl_THUN_12 genome, and (C) the *B.g. tritici* Bgt_CHE_96224 genome. **Fig G. BUSCO results of the pangenome.** BUSCO analyses results for all the wheat and rye powdery mildew genomes that have been long read sequenced with PacBio and assembled into chromosomes on three different levels of databases being used: (A) Fungi, (B) Ascomycota, and (C) Leotiomycetes. **Fig H. BUSCO results of the pangenome's CDS.** BUSCO analyses results for all the wheat and rye powdery mildew coding sequences (CDS) that have been analysed with maker using the near-chromosome scale assemblies, on three different levels of databases being used: (A) Fungi, (B) Ascomycota, and (C) Leotiomycetes. **Fig I. Gene annotation statistics.** (A) Number of effector and non-effector genes for all the isolates in the pangenome. Grouped effectors refers to effectors that are homologous to a known effector family, while the ungrouped ones do not belong to a known effector family. (B) Distribution of size of proteins for all the proteins for each isolate. **Fig J. Genome statistics and phylogeny of the pangenome.** Chromosomes sizes for all isolates. **Fig K. Synteny of *B. graminis* and *B. hordei*.** (A) Synteny of *B.g. tritici* Bgt_CHE_96224 (all 11 chromosomes) with the nine largest contigs of *B. hordei* Bh_DH14 using 10kb windows of minimum alignment. (B) Same as (A) with the 100

largest contigs of *B. hordei* Bh_DH14 instead. (C) Dot plot comparison of chromosome 11 from Bgt_CHE_96224 and *B. hordei* Bh_DH14. Chromosome 11 of *B. hordei* was assembled from three sequences contigs [23,26,33] based on their best blast homologies to chr11 of Bgt_CHE_96224, which is why it was not represented in A. Note that sequence homology and collinearity on the left arm is well visible, while it is barely detectable on the right arm. **Fig L. Genome statistics and phylogeny of the pangenome.** PhyloNet network on splitstree using single-copy orthogroups via Orthofinder with 100 bootstrap (some of the values are missing from the figure). **Fig M. Pangenome graph for the 11 isolates included in the *Blumeria graminis* pangenome. Fig N. Mitochondrial genome visualisation and synteny.** (A) Mitochondrial genome of isolate Bgt_CHN_17_40 along with its annotation of genes etc. The inner circle represents the GC content of the mitochondrial genome. (B) Synteny between the mitochondrial genome of Bgt_CHN_17_40 and Bh_DH14. **Fig O. Pairs of whole genome synteny for various genomes.** (A) Syntenic pair of the Bgs_1459 and the Bgt_CHN_52_27 genomes in 10kb search windows of synteny, (B) Syntenic pair of the Bgs_1459 and the Bgt_JPN_CHIKA genome in 10kb search windows of synteny, (C) Syntenic pair of the Bgs_1459 and the Bgt_USA_2 genome in 10kb search windows of synteny, (D) Syntenic pair of the Bgt_CHN_52_27 and the Bgt_USA_2 genome in 10kb search windows of synteny. The asterisk represents the genome that is a hybrid one. **Fig P. Heatmap of independent chromosome average nucleotide identity (ANI) comparisons, using OrthoANIu tool.** (A) Chr-01, (B) Chr-02, (C) Chr-03, (D) Chr-04, (E) Chr-05, (F) Chr-06, (G) Chr-07, (H) Chr-08, (I) Chr-09, (J) Chr-10, (K) Chr-11, (L) Boxplot of the ANI values for chromosome pairs (with – right side and without Bgs – left side) for all chr from 01 to 10, and then only for chr11. A Kruskal-Wallis rank sum test was performed. The p-values show statistically significant difference between chr11 pairs and all the rest of the chromosomes in pairs ANI values in both cases (including & excluding the Bgs chromosomes). **Fig Q. Sequence conservation and diversity along the chromosomes.** Sequence conservation across all chromosomes from chr-01 to chr-11. The heat maps show comparisons of *B. graminis* isolates with the reference isolate Bgt_CHE_96224 in 50 kb windows on the top. The black vertical bars on top signify the non-effector genes, while the red vertical bars below the black ones signify the effector genes. The red line in the middle shows the average sequence conservation for all 50kb windows among *B.g. tritici* isolates, while the gray line shows sequence conservation between Bgt_CHE_96224 and Bgs_1459. Note that sequence conservation is generally lower in the broader centromere. Also, there are a few small regions in a few chromosomes (chr-01, chr-02), where the sequence conservation is much lower than the rest of the chromosome. The ratio of the sequence conservation between *B.g. tritici* isolates and Bgs_1459 is shown at the bottom of the subfigure in pink. **Fig R. Dot plot comparisons of chr-11 of *B. graminis* isolates.** (A) Comparison of chr-11 from the *B.g. tritici* isolates Bgt_CHE_96224 and Bgt_CHN_17_40. Note that Bgt_CHN_17_40 is missing segments of ~180 kb and ~150 kb at the left and right ends, respectively. (B) Comparison of chr-11 form the *B. g. tritici* isolates Bgt_CHE_96224 and Bgt_ARG_4_2. Note that Bgt_ARG_4_2 has an ~150 kb segment at the left end that is not found in Bgt_CHE_96224 and other *B.g. tritici* isolates. **Fig S. Gene presence/absence polymorphisms along *B. graminis* chromosomes detected through sequence coverage with Illumina reads in 399 *B. graminis* isolates.** (A) Copy number variation (CNV) along chromosomes. The x-axis indicates the position in Mb while the y-axis shows the number of *B. graminis* isolates that show copy number variation in a given gene. Circles indicate multiple copies of genes while triangles indicate deletions. Genes were separated into effectors (red) and non-effector genes (gray). Centromeres can be recognized as gene-free regions. (B) Summary of proportions of genes that show CNV. Effectors and non-effectors are shown separately, as are values for all chromosomes and for the right arm of chromosome 11. The y-axis shows the percentage of genes that show duplications and/or deletions in at least 1 isolate. Note that the total of duplications and deletions can exceed 100% because some genes may be duplicated in one isolate but deleted in another. **Fig T. Effector protein phylogeny.** Phylogenetic tree of all candidate effector proteins from the right chromosomal arm for Bgs_1459, Bgd_ISR_211, and Bgt_CHE_96224, along with the effectors on the left start region that is missing in the *B.g. tritici* isolates and homologous effectors of Bgs1459–09640 found in chr-06. We used Bayesian inference with MrBayes on ClustalW aligned amino acid sequences of all and the outgroup candidate effector Bgs1459–00747. The effectors in black font colour refer to all the effectors of the right chromosomal arm of chr-11, the ones in red colour refer to all the effectors on

the left start region which is missing in *B.g. tritici* isolates and the ones in green refer to the effectors in chr-06. The outgroup effector has an asterisk on the right side. **Fig U. Candidate effector comparisons for proteins: BgTH12–07698 and Bgs1459–09656.** (A) Annotation and alignment of RNAseq reads of the BgtI_THUN_12 isolate infecting the triticale culitvar Timbo at two days post inoculation that was used to verify the expression and correct annotation of two candidate effector genes of interest. (B) Alignment of the unique candidate effectors found only in the *B.g. secalis* and *B.g. triticale* isolates, along with the closest homologous other candidate effector protein found in all isolates in chromosome 7. Cysteines for possible cysteine bridges at positions 30 and 66. (C) Genealogic tree of candidate effector genes that had at least some homology with the candidate effectors above. Most of these proteins belong to the effector family E003. BgtE-20034 protein was used as an outgroup. **Fig V. Establishment of thresholds to determine gene copy numbers based on Illumina sequence read coverage.** Read coverage was calculated for individual genes across all 399 mildew isolates used in this study. For each gene, it was statistically tested whether reads coverage across the 399 isolates showed uni-modal or bi-modal distribution. (A) Read coverage distribution for genes that showed uni-modal distribution. As threshold for multiple copies, the upper 1 percentile was used (1.7, indicated in red), meaning that 99% genes have read coverage below 1.7. (B) Read coverage distribution for genes that showed bi-modal distribution. Since the group of genes without coverage was very distinct, a value of 0.1 was used (indicated in red), more stringent than the value of the lowest percentile (0.409 shown in A). **Fig W. Duplications and deletions in various datasets.** Duplications and deletions for all isolates or subsets of isolates per population for most geographic populations and for all genes (see Table B in S2 Appendix). (A) Duplications in subsets of eight randomly chosen isolates per population, (B) Duplications in all isolates per population, (C) Deletions in subsets of eight randomly chosen isolates per population, (D) Deletions in all isolates per population. **Fig X. Heatmap of CNVs of effector genes in various isolates.** The Bgt_CHE_96224 isolate has a consistent blue colour of medium dark blue for all genes. Darker blue colour than this signifies a possible deletion of a gene, while lighter blue to red colours signifies possible different levels of duplication of a gene. Note that despite its normalization for read coverage and against the reference, many arte-facts might still exist due to annotation biases, searching biases etc. Each column refers to one isolate, while each row refers to one gene. (A) refers to the effector genes, while (B) refers to non-effector genes. **Fig Y. Phylogenetic network of one of the most diverse effector families in wheat powdery mildew (E014). Fig Z. Examples for effector gene presence/ absence polymorphisms caused by unequal homologous recombination.** The left side of the panels shows a dot-plot alignment of the region carrying the polymorphism. Gene and TE annotation for the mildew genome that carries additional sequence is shown above the plot with transcriptional orientation indicated by arrow heads. For easier visibility, the TE or gene that served as template for the unequal homologous recombination is connected by shaded areas to the corresponding point in the plot. The right side of the panel shows a multiple alignment of the sequence that serves as template. Recombi-nant sequences (which is found in the isolate that carries the deletion) has the extension "rec". Note that the rec sequence, as expected, shares diagnostic SNPs with the left template in the 5' region and with the right template in the 3' region, respectively. **Fig AA. TE analyses in *Blumeria graminis*.** TE counts of TE superfamilies (and other TEs) for the pange-nome. **Fig AB. Genome wide distribution of various TE families of interest.** Distribution of various TE families across the Bgt_CHE_96224 genome after a blast with a cutoff of hits >500 bp and with 70% identity. (A) *RII_Amitabha_C*, (B) *RII_Aves*, and (C) *RII_Itera*.
(PDF)

**S2 Appendix. Table A.** List of isolates used in one or more analyses and details about them. The forma specialis is deter-mined based mostly on the genomic PCA, as in some cases the type of host is not available. **Table B.** List of eight random isolates used for multiple analyses (1st column) (randomised using random.org). Singletons results for the random eight isolates per population. **Table C.** List of isolates used in the Mantel test analyses. The *B.g. dicocci* isolates are completely excluded from these analyses. **Table D.** CRAQ (Clipping Reveals Assembly Quality) results for reference-free genome assembly evaluation including the accuracy of assembled genomic sequences for the isolates that were sequenced in this study. **Table E.** Statistics of genomes in *Blumeria graminis* pangenome. **Table F.** Pangenome statistics including

number of nodes, edges, and the total length of the pangenome. **Table G.** Whole genome average nucleotide identity (ANI) comparisons, using OrthoANIu tool. Also, mitochondrial genomes are compared, along with genes (CDS). **Table H.** Number of genes/ORFs in each mitochondrial genome, under two annotation pipelines. **Table I.** Mating types in the *B. graminis* pangenome. We blasted the genomes to look for the mating type genes/loci. *MAT 1-1-1* corresponds to the gene *BgtCHN52_27_00382* in isolate Bgt_CHN_52_27, *MAT 1-2-1* corresponds to the gene *Bgt-3306* and *SLA-2* to the gene *Bgt-2805* in isolate Bgt_CHE_96224. **Table J.** Presence or absence of the characterized *AvrPm2* gene *BgtE-5845* coming from Bgt_CHE_96224, along with two other genes of the same effector family in the long-read assemblies of the pangenome isolates. The percentage represents the DNA homology between the *BgtE-5845* gene that was used as a query in the blast analysis and the assemblies, if no percentage present, then it implies 100% identity. **Table K.** TE statistics of the B. graminis pangenome using only copies of identified and annotated TEs, using the genome size, and averaging for the size of identified TEs for the various types (DNA TEs, LINEs, SINEs, LTRs). **Table L.** List of isolates and populations used for the detettore analyses. **Table M.** Number of copies of the TE RLX_Bgrt_RXB_consenus-1 among the various genomes in the pangenome dataset. **Table N.** Statistics of *Blumeria graminis* genome assemblies initial sequencing output. **Table O.** Statistics of *Blumeria graminis* genome assemblies after assembly using the software gaas. **Table P.** Details on the transposable element identification on the Bgt_CHE_96224 isolate and beyond. (XLSX)

## Acknowledgments

The authors would like to acknowledge Dr. Coraline Praz and Dr. Manuel Amos Poretti for the discussions with the authors regarding powdery mildew transcriptomics. The authors would like to acknowledge the Traditional Custodians of the Toowoomba, Magandjin/Meanjin and surrounding regions (including the Giabal, Jarowair, Yuggera and Turrbal Peoples).

## Author contributions

**Conceptualization:** Alexandros G. Sotiropoulos, Beat Keller, Thomas Wicker.

**Data curation:** Alexandros G. Sotiropoulos, Marion C. Müller, Lukas Kunz, Edith Schlagenhauf.

**Formal analysis:** Alexandros G. Sotiropoulos, Marion C. Müller, Lukas Kunz, Johannes P. Graf, Thomas Wicker.

**Funding acquisition:** Levente Kiss, Beat Keller, Thomas Wicker.

**Investigation:** Alexandros G. Sotiropoulos, Marion C. Müller, Lukas Kunz, Johannes P. Graf, Thomas Wicker.

**Methodology:** Alexandros G. Sotiropoulos, Marion C. Müller, Lukas Kunz, Johannes P. Graf, Edith Schlagenhauf, Thomas Wicker.

**Project administration:** Alexandros G. Sotiropoulos.

**Resources:** Alexandros G. Sotiropoulos, Lukas Kunz.

**Software:** Alexandros G. Sotiropoulos, Marion C. Müller, Johannes P. Graf, Edith Schlagenhauf, Thomas Wicker.

**Visualization:** Alexandros G. Sotiropoulos, Marion C. Müller, Thomas Wicker.

**Writing – original draft:** Alexandros G. Sotiropoulos, Thomas Wicker.

**Writing – review & editing:** Alexandros G. Sotiropoulos, Marion C. Müller, Lukas Kunz, Levente Kiss, Ralph Hückelhoven, Edith Schlagenhauf, Beat Keller, Thomas Wicker.

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
