## [Decision Letter · Decision Letter 0]

22 Jun 2025

Combining pangenomics and population genetics finds chromosomal re-arrangements, accessory-like chromosome segments, copy number variations and transposon polymorphisms in wheat and rye powdery mildew

PLOS Pathogens

Dear Dr. Sotiropoulos,

Thank you for submitting your manuscript to PLOS Pathogens. After careful consideration, and based on the reviewers' comments, we feel that it has merit but does not meet PLOS Pathogens's publication criteria as it currently stands. In particular, reviewer 2 and 3 highlight additional analyses that are necessary to further strengthen the submission (e.g., pop. genomic analyses, TEs, accessory chromosomes), as well as the description of the pan-genome analyses. Moreover, reviewer 3 highlight that the manuscript needs to be revised to highlight novel findings rather than reinforcing previous work. Therefore, we invite you to submit a revised version of the manuscript that addresses all the points raised during the review process.

Please submit your revised manuscript within 60 days Aug 21 2025 11:59PM. If you will need more time than this to complete your revisions, please reply to this message or contact the journal office at plospathogens@plos.org. Please include the following items when submitting your revised manuscript:

We look forward to receiving your revised manuscript.

Kind regards,

Michael F. Seidl

Academic Editor

PLOS Pathogens

Debra Bessen

Section Editor

Editor-in-Chief

PLOS Pathogens

PLOS Pathogens

orcid.org/0000-0002-7699-2064

**Journal Requirements:**

1) We have noticed that you have uploaded Supporting Information files, but you have not included a list of legends. Please add a full list of legends for your Supporting Information files after the references list.

2) Some material included in your submission may be copyrighted. According to PLOSu2019s copyright policy, authors who use figures or other material (e.g., graphics, clipart, maps) from another author or copyright holder must demonstrate or obtain permission to publish this material under the Creative Commons Attribution 4.0 International (CC BY 4.0) License used by PLOS journals. Please closely review the details of PLOSu2019s copyright requirements here: PLOS Licenses and Copyright. If you need to request permissions from a copyright holder, you may use PLOS's Copyright Content Permission form.

Potential Copyright Issues:

i) Figure 1. Please (a) provide a direct link to the base layer of the map (i.e., the country or region border shape) and ensure this is also included in the figure legend; and (b) provide a link to the terms of use / license information for the base layer image or shapefile. We cannot publish proprietary or copyrighted maps (e.g. Google Maps, Mapquest) and the terms of use for your map base layer must be compatible with our CC BY 4.0 license.

3) Please amend your detailed Financial Disclosure statement. This is published with the article. It must therefore be completed in full sentences and contain the exact wording you wish to be published.

1) State the initials, alongside each funding source, of each author to receive each grant. For example: "This work was supported by the National Institutes of Health (####### to AM; ###### to CJ) and the National Science Foundation (###### to AM).".

**Reviewers' Comments:**

Reviewer's Responses to Questions

**Part I - Summary**

Reviewer #1: The authors conducted a comparative genomics and pangenome analysis involving eight hexaploid wheat powdery mildew (Blumeria graminis f.sp. tritici), a tetraploid wheat powdery mildew, a species specialized to infect rye and one for triticale. They provide seven new high-quality genome assemblies and included published genomes as well as sequencing datasets of 399 isolates. Their analysis reveals chromosomal rearrangements within several of these species and/or isolates and highlights a near-accessory quality of chr-11, which was previously unseen for powdery mildew fungi. They further enriched the analysis and found extensive copy number variation among effector groups as well as signs of recent transposable element activity, evidence for which have previously been presented for both wheat and barley powdery mildew. Their work is the first pangenomic analysis of powdery mildew fungi and as such relevant, since such analysis are challenging with obligate biotrophic plant pathogens.

Reviewer #2: This study by Sotiropoulos et al. presents a valuable set of 11 high-quality Blumeria graminis genome assemblies from globally distributed isolates, offering a useful resource for fungal genomics and plant-pathogen interaction research. While the sequencing and assembly work is technically strong, the manuscript would benefit from clearer terminology and structure, particularly regarding the use of the term “pangenome,” which is not supported by formal construction of a pangenome graph. Key interpretations, such as the classification of chr11 as "accessory-like" and the analysis of TE content, could benefit from more cautious framing and deeper analysis. The reuse of previously published data without clearly highlighting new insights also reduces clarity. Nonetheless, the data generated are of high value, and with improved organization, clearer distinctions between reused and novel findings, and more rigorous comparative analyses, the study could significantly contribute to the understanding of genome evolution in this species complex.

Reviewer #3: This large-scale study uses a combination of long-read whole genome sequencing from 9 Pgt isolates and two additional isolates specialized on other hosts as well as population level and short read sequencing of 399 isolates to explore the pangenome of this important obligate biotroph. This work has one of the largest number of isolates sequenced (~400) second only to Z. tritici (1000). Using these data the authors explore whole genome synteny, within population diversity, TE diversity and insertion polymorphisms. While I think that the data and analysis supporting this work is very strong, I found that overall, main emphasis in the results and discussion simply reinforces prior studies on this species complex (i.e. populations are more diverse at the centre of origin and TE insertions are also greater). I feel that the text could be substantially revised to place more emphasis on the new insights that these long-read genomes brought to this important species complex and how the population level data can be used to re-enforce the findings from these long-read assemblies (i.e. effector gene copy number variation in the accessory chr11, can you show the duplicated genes in the context of the chromosome? And then validate/show variation for a large number of isolates from that same population?).

I note the large number of supplemental figures and tables. Unfortunately, as a reviewer I could not access the supplementary tables. Given the large amount of data generated I would encourage the authors to submit these tables-data this to Zenodo to increase accessibility.

My comments below highlight interesting aspects of the work, which I feel could be further clarified or expanded upon to emphasise some of the novel findings of this work.

**Part II – Major Issues: Key Experiments Required for Acceptance**

Please use this section to detail the key new experiments or modifications of existing experiments that should be absolutely required to validate study conclusions.required to validate study conclusions.required to validate study conclusions.required to validate study conclusions.

Reviewer #1: (No Response)

Reviewer #2: Major Comments

1. Definition and use of the pangenome concept

The manuscript would benefit from a more precise and accurate definition of the term “pangenome.” The authors use this term to describe a collection of reference genomes, but they do not construct a pangenome graph. A true pangenome typically involves a structured representation (e.g., a gene presence/absence matrix or a graph-based approach) that systematically captures shared and variable genomic content across strains. The manuscript should consistently differentiate between the comparison of multi-reference genomes and an actual pangenome. I recommend revising the terminology throughout the manuscript to reflect this distinction.

2. Structure and flow of the manuscript

The manuscript currently lacks clear structure and logical flow, which makes it difficult to follow. It is often unclear what constitutes novel findings versus what has been confirmed or re-analyzed from previous work. Greater effort is needed throughout to clearly delineate new data from reused material.

3. New resources data

The first results section titled “New resources for the worldwide study of wheat powdery mildew” reuses published population data and confirms previously reported findings. This is not a problem per se but I struggled to get the novelty. If the aim is to justify the selection of strains for long-read sequencing, this could be presented more concisely as rationale rather than as a primary result. This section could be moved to the supplementary material (with Figure 1 as Figure S1). If the authors wish to highlight new insights into population structure, they should consider complementing the PCA with an ADMIXTURE analysis. This could clarify whether the Central European, Middle Eastern, and Russian samples form a single cluster or exhibit more nuanced structure.

4. Mitochondrial genome assembly

Is the mitochondrial genome of Bgt_CHN the only one that is fully assembled? What was the rationale for including the mitochondrial genome + unassembled contigs as chr_Un in the other assemblies instead of extracting the mitochondrial genome directly using Bgt_CHN as a reference? This needs clarification, as the current approach appears inconsistent. In addition, I would prefer to retain the unassembled contigs as they are, rather than attempting to artificially assemble them into a chimeric 'chromosome' that lacks biological meaning.

5. Chromosome 11 with characteristic of accessory chromosome

While the authors describe chr11 as having "features of accessory chromosomes," I believe this framing is misleading and detracts from the more biologically meaningful interpretation. Namely, that chr11 may reflect a history of introgression. There is no evidence of presence/absence variation across strains, which is fundamental to defining accessory chromosomes. In fungal genomics, including cases like Zymoseptoria tritici, core chromosomes can undergo significant rearrangements, harbor accessory genes, or result from introgression without being classified as accessory. The emphasis should be placed on the signals of introgression and their evolutionary implications, rather than on drawing parallels with accessory chromosomes, which may not be warranted in this case.

6. TE content analysis

The comparison of TE content between reference genomes could be expanded. Are there shared TE families across strains? A clustering approach could identify TE families conserved among strains or specific. This analysis would be particularly insightful for hybrid strains (e.g., the Japanese isolate), potentially revealing "parent"-specific TE inheritance. Additionally, identification of TE families found in the multi-reference context would highlight the added value of this dataset over single-reference approaches as used to detect TIPs.

Reviewer #3:

For a pan-genome study I found it odd that the authors did not explicitly report or provide a summary or overview of the pangenome size (i.e. what proportion of genes are found in all isolates, core vs accessory genome, and given the discovery of chr11 what proportion of accessory genes are on this chr both within and between forma specialis).

Better integration with existing literature on TEs and fungal pangenomes. Throughout the intro and discussion there are several opportunities to better incorporate existing literature from other pathosystems, for example the long-read Myrtle rust genomes which also have exceptionally high TE content and are also obligate biotrophs. Similarly, the discussion would be strengthened by placing this work into context with other fungal pan-genome studies, for example Fusarium, Pyrenophora tritici-repentis, Verticillium…

Copy number variation in pangenome isolates vs. long-read assemblies. Given the authors had some examples of potentially very recent gene duplication events (i.e. E254n in Bgs1459-09640 found in chr-06 and E003 in multiple isolates in chr11, shown in FigS13) I feel it would have been much more informative to look at the location of these genes in the long-read assemblies in more detail, rather than simply report the relatedness of the broader effector family in a phylogenetic tree (Fig5D). The findings of effector gene copy number variation matches reports from other pathosystems, Rhyncosporium secalis, however this study has the added advantage of the long-read assemblies to dig into these copy number variations. For example, how are these extra copies arranged on the chromosome in the long-read assemblies, in-tandem? Do the surrounding genome feature give any hints as to the mechanism driving gene duplication (TE capture, unequal crossing over). Figure 5C and D focuses on very large effector families, but it was unclear to me here within an effector family (E014) which genes had copy number variations.

Line 224-226: Genome size differences were driven largely by “repetitive regions in the centromeres”…where was this data shown-summarised? Figure S3 does not detail the size of the changes, just shows a heat-map of the transposon location and nucleotide (?) similarity in the heatmap legend.

Throughout the results and discussion there is reference to a “hybrid” Japanese population with reference to earlier studies. However, in this manuscript this is not given any introduction and therefore makes the reference to this prior work confusing and difficult to interpret.

Figure 1C: Authors state that nucleotide diversity was measured in 10 kb windows for eight randomly chosen isolates from populations with at least eight isolates. Should the data identifying the isolates chosen be provided in a supplementary table? (or are these the isolates shown in FigureS2a, if yes this should be stated clearly in the text. Did the authors perform this analysis more than once with different individuals to see if this influenced the results? The ANOVA presented seems to show pair-wise p-values for populations drawn next to each other in the graph, however this does not give a sense of groups (i.e. is the Nuc diversity in Bgdic significantly lower than the USA or ARG?). Would not a more traditional ab abc bc notation make this analyses more clear?

Figure 1D: What do the blue, yellow and red colors represent in the background of this plot? Density of data points?

Figure 2: “except in the highly repetitive centromeric regions” Figure 2 does not show any genomic features that would allow us to assess if synteny breaks are around centromeric sequences.

Figure 3B and accompanying text 304-315: I did not quite understand the relevance of this section which refers to putative misassemblies. The author simply state here that they looked re-arrangements close to assembly gaps. However, a much more thorough way to investigate this would be to align your PacBio reads back to your assembly and see if you have reads that span the breakpoints of the translocations. This is easily done with minimap2 or similar software.

Figure 4E-H: Given the termini of Chr11 were highly variable both within and between f.s. why were the genes in these regions not investigated in more detail? Did these regions end up in the un-assigned contigs? Or are they truly absent?

Maybe I missed the download link but only PDF with Supp Figures and Supp dataset 2.xls was available to reviewers. I was unable to review any of the ten supplementary tables, please consider Zenodo.

**Part III – Minor Issues: Editorial and Data Presentation Modifications**

Reviewer #1: Minor analytical/experimental issues:

1. L268ff: mitochondrial genome - is the gene content complete for a fungal mitochondrial genome? Refer to 10.1038/s41598-021-93481-5 and 10.1099/mgen.0.000720 for further information of mitochondrial genomes in powdery mildews.

2. L304ff: rearrangements. The authors exclusively searched for sequence gaps to identify potential breakpoints. However, much stronger evidence for misassembly errors are to use remapping inconsistencies between the Pacbio reads and the respective assemblies (e.g. with a tool such as CRAQ, which also assesses genome assembly quality and reports read/assembly agreement both locally and globally). In addition, did the authors consider using experimental evidence via chromatin capture-based methods (HiC or PoreC) to correctly arrange all contigs?

3. L699ff (material and methods; manual contig scaffolding): If I understand correctly, the authors manually scaffolded the contigs based on the B. graminis f.sp. tritici CHE_96224 reference assembly and introduced arbitrary stretches of N between them (or did they use an automated process such as RagTag? not described). Assembling the scaffolds based on a reference genome risks introducing reference bias in the genome assemblies. How did the authors account or correct for this potential bias?

4. L321ff, chr-11 shows characteristics of an accessory chromosome. Did the authors test if any B. hordei DH14 or RACE1 contigs match chr-11, to estimate if this chr-11 is specifically present within the B. graminis lineages or potentially shared within the Blumeria genus?

5. L388: the authors mention that one gene was found only in B. secalis and B.g. triticale. What did this gene code for?

6. L494ff, TE distance analysis: this analysis is a little unclear to me, please clarify. In particular, is the analysis limited to TEs adjacent to genes and at what distance cut-off? what are the 21000 TEs on average referring to, those neighboring genes? I think the description would be clearer if it would describe TE distance from the gene point of view, given the TE-enriched nature of the powdery mildew genomes making up the majority of the sequence.

7. It occurs to me that LINEs and SINEs were also among the well-expressed TEs in B. hordei (L510ff). Did the authors use RNA-seq data (public or other) to check if these LINE and SINE elements also exhibit significant expression levels, coinciding with their observed genomic mobility?

8. Methodology: variant calling (L660ff): based on the parameters, it seems there was no filtering by allele frequency (AF vs RF). This may confound the SNV analysis since the variant callers tend to call many SNVs even if the alternative allele frequency is well below 0.8. Did the authors filter by alternative allele frequency in some way?

9. L674ff, population genomics analysis: I have tried to understand how this analysis is done, but this section only lists the names of R packages. Please describe the approach used for the population genomics, and/or refer to a public repository with a detailed code collection, so the method can be replicated.

10. L731ff: TE identification. The authors used manual curation and EDTA to create a TE library. How was genome-wide TE detection done, with RepeatMasker or differently?

L812-814: the coverage cut-off was 0.1-1.7 (above 1.7 was considered duplicated). Did the authors also consider higher coverage and thus higher level copy number variations? what were the cut-offs for e.g. three, four, five, etc. copies?

Minor typographic issues:

L149: 'about 399 isolates', remove 'about'

L166/167: the reads were mapped to the genome or assembly, not the isolate itself. Rephrase.

L212ff: which eight B. graminis isolates exactly were sequenced? Please explicitly mention here (also L694).

L270: remove the word 'us' between 'showed high levels of...'

L410: change to 'possible underestimation'

L455: correct 'Transposable Elements insertions' to 'Transposable element insertions'

L506: comma before which missing

L507: 'posing a known bias', which bias? Provide a reference

L508-509: something... this statement is very vague and unclear, please rephrase into a precise and clear conclusion.

L517: reference missing for this statement

L563: replace 'from' with 'by'

L569: 'highest number of variation' do the authors mean 'level'?

L586-587, 'couple less Ma' rephrase into a less colloquial statement

L592, 'which is puzzling': colloquial phrase, remove

L591ff, TE proximity upstream of genes: TEs can also act as promoters for genes, thus increasing expression

L597: CITE 4 papers, I believe references are missing here

Reviewer #2: Minor Comments

Line 74: The term “Earlier…” is ambiguous. Please clarify, do you mean earlier in the literature, or earlier in this manuscript?

Line 136: The phrase “addition of genes not found by short reads” needs elaboration. Are the authors referring to limitations in short-read based assemblies, or to gene presence/absence undetectable by resequencing? or something else?

Line 137: Please explain how using multiple reference genomes improves gene expression analysis compared to a single reference.

Line 138: Clarify how “host pangenomes” are relevant to Blumeria genomics.

Line 166: Please include citations in Table S1 to support all referenced datasets.

Line 195: “statistically less diverse mean” is unclear. Also, was the low diversity in the Australian population due to limited sampling? If PCA shows USA and Australian populations overlap, are they truly isolated?

Lines 201–202: The phrase “as hypothesised before” is more appropriate for the discussion. The results section reads as a re-confirmation of previous findings, raising concerns about novelty.

Lines 204–209: Clarify how geographic groups were defined for the Mantel test. Explicitly state conclusions about spatial vs genetic distance correlations. Could trade explain the observed patterns?

Figure 1:

In 1A, the triangle symbol for B. graminis dicocci is indistinct.

In 1B, are both dicocci and tritici isolates shown? Do formae speciales cluster separately? What does this imply about host specificity?

In 1C, clarify whether the p-values shown represent all or only significant pairwise comparisons. The figure legend is unclear. What does “based on 8 randomly chosen isolates” mean?

Table S1: Please include references for each dataset. Were any new Illumina reads generated for this study? This was not clearly stated.

Figure S6: Define “grouped” vs. “ungrouped” effectors. Do you mean effector multigene families?

Figure S7A: Consider ordering the x-axis by chromosome number for readability.

Line 220: Using the term “pangenome” is misleading if a pangenome graph was not created. Please clarify.

Line 226: Are centromeric repeats the sole drivers of genome size variation? How was this determined?

Line 239: Was chr_Un generated by merging unassembled contigs? If so, this may be misleading. Unplaced contigs should remain as such rather than being artificially grouped.

Line 240: Why is part of the mitochondrial genome found in chr_Un? Normally, mitochondrial genomes are assembled and analyzed separately.

Line 245: The phrase “which shows the potential indication of recombination” is vague. What precisely indicates recombination here?

Line 246: What observation is being referred to regarding Bgt_JPN? Please clarify.

Lines 273–279: Are these loci consistently located at the same positions across all assembled genomes? While I would expect them to be conserved, it would be interesting to know whether any structural rearrangements have occurred in the regions surrounding the mating-type loci. Given the potential implications for recombination and reproductive isolation, this could be an informative aspect to address, particularly in the context of genome structure variation.

Lines 306–307: Clarify which rearrangement is being referenced.

Line 400: It is unclear which reference genome was used to assess copy number variations (CNVs). Did the authors use a single reference genome for all populations, or were population-specific references employed? If only one reference was used, it raises the question of how leveraging multiple high-quality genomes in a pangenome framework adds value to the CNV analysis. A key advantage of a pangenome approach is the ability to reduce reference bias and better capture structural variation across diverse lineages—this benefit would be diminished if CNVs are mapped solely to a single genome.

Line 448: Is AvrPm2 present in all isolates? Is there any host specialization linked to its presence or absence?

Line 520: Was this TE family with high number of TIPs in the population, also depicts a high TE copy number in the TE annotation of the reference assembly of Bg dicocci?

Line 542: The statement linking accessory chromosomes to virulence in Z. tritici is misleading and should be revised. In this species, accessory chromosomes are not directly associated with virulence. In fact, studies have shown that the loss of certain accessory chromosomes can even enhance virulence on specific hosts (e.g., Habig et al., 2017).

Line 567: The wide variation in chr11 size and introgression suggests large-scale structural rearrangement, contrary to this sentence stating of no major rearrangements between Bg genomes.

Lines 591–592: The observed proximity of TEs to effector genes is a well-documented feature in many fungal plant pathogens. However, the authors should also acknowledge that overlap between TE and effector annotations can sometimes result from annotation artifacts, particularly in repeat-rich genomes. It would be prudent to mention that such cases may require manual curation to distinguish true gene–TE associations from misannotations.

Line 597: Please replace “(CITE 4 paper)” with the actual references.

Line 612: The authors may wish to include a brief discussion contrasting pangenome graphs with the multi-reference genome approach used in this study. While the availability of multiple high-quality assemblies is undoubtedly valuable, a pangenome graph offers a structured framework for representing gene content and structural variation across strains, enabling more systematic comparative analyses. Discussing the trade-offs between these approaches (such as resolution, computational complexity, and biological interpretability) would help clarify the scope and limitations of the current work, and situate it more clearly within the broader context of pangenomic research.

Reviewer #3: The authors report a high level of synteny between the long-read genome assemblies. In Fig3 several large-scale rearrangements are reported that contain dozens of genes. Did the authors investigate what genes these were and/or the flanking regions to better understand 1) what may have driven the rearrangement 2) do these genes appear to have important functions that would require co-regulation (secondary metabolites).

Lines 119-120 and 127-128:There are some gaps in the introduction in terms of recent publications from many rust fungi (also an obligate biotroph) that have also been shown to have very high TE content (https://doi.org/10.1093/g3journal/jkaa015).

Line 207: “broader region of origin” can you be more explicit in this sentence about which populations you included here (“Middle Asia” in Figure 1D)

Line 216 and Fig1A and B: It would be helpful to know where isolate THUN-12 is from? Why are ISR_7 and CHE-96624 highlighted in the map but not this other isolate? Also there are nine names highlighted in Figure 1A, only eight names highlighted in Figure1B but eleven isolates sequenced with long reads (two B.g. dicocci names not shown?). Please revise to be consistent.

Line 217: Please name country of origin for B. hordei isolate.

Line 246: why was Bgt_JPN_CHIKA expected to be an outgroup with the other isolates in Figure S7? It appears to group within the main cluster of Bgt isolates, so this sentence does not make sense. Also in this figure what do the different colors mean? Population of isolation?

Lines 255-256: The referral to B hordei DH14 in the figure legend for Figure2 appears slightly out of context. Is this the most appropriate place for this? There is not accompanying text in the main results for this S8 figure.

Line 259-262: run on sentence, please revise.

Line 279: Mating type analysis, can you please detail which isolates belong to which mating type

Line 284-289: This section of the results refers to some previous published (?, ref7?) knowledge of the Japanese population being a hybrid of a US and CHN isolate. It would be helpful if this could be more thoroughly introduced or placed into context before discussing these genome translocations.

Line 552-553: “our work has supported” this sentence does not give details as to which data-results from this work supports the use of the new naming nomenclature, it would be good to be more explicit here and refer to supporting data (i.e. whole genome synteny and accessory chr11)

Line 593-597: I was surprised that the authors refer to the close association of effector genes to TEs as “puzzling”, given this has been reported for many fungal species. Please see https://www.sciencedirect.com/science/article/abs/pii/S0168952521002298 for a comprehensive review. Also this section concludes with (CITE 4 papers).

Pangenome analyses of many pathogenic fungi has also been performed: see

Linked to this is a suggestion that the Discussion could certainly be improved and made more accessible to the wider evolutionary biology community by discussing the TE content of these fungi in the context of their obligate biotrophic lifestyle.

PLOS authors have the option to publish the peer review history of their article (what does this mean?). If published, this will include your full peer review and any attached files.). If published, this will include your full peer review and any attached files.). If published, this will include your full peer review and any attached files.). If published, this will include your full peer review and any attached files.

...

Reviewer #1: **Yes:** Stefan KuschStefan KuschStefan KuschStefan Kusch

Reviewer #2: No

Reviewer #3: No

**Figure resubmission:**

**Reproducibility:**



---

## [Decision Letter · Decision Letter 1]

2 Dec 2025

Combining pangenomics and population genetics finds chromosomal re-arrangements, diversified chromosome segments, copy number variations and transposon polymorphisms in wheat and rye powdery mildew

PLOS Pathogens

Dear Dr. Sotiropoulos,

Thank you for submitting your manuscript to PLOS Pathogens. Your manuscript has been reviewed by the initial reviewers. While two of the three are satisfied with the adjustments to the manuscript, one of the reviewer raises several concerns that need to be addressed prior acceptance. In particular, the availability of the code and details to reproduce the bioinformatic analyses are essential, relating to several comments (see below). Moreover, I concur with reviewer 3 that the manuscript should include a basic description of the pan-genome as well as a careful consideration of Chr-Un. After careful consideration, we therefore feel that it currently does not yet fully meet PLOS Pathogens's publication criteria as it currently stands. Therefore, we invite you to submit a revised version of the manuscript that addresses the points raised during the review process.

We look forward to receiving your revised manuscript.

Kind regards,

Michael F. Seidl

Academic Editor

PLOS Pathogens

Debra Bessen

Section Editor

PLOS Pathogens

Editor-in-Chief

PLOS Pathogens

orcid.org/0000-0003-2946-9497

Editor-in-Chief

PLOS Pathogens

orcid.org/0000-0002-7699-2064

**Journal Requirements:**

1) Thank you for stating "Raw PacBio and Illumina sequences are available in the SRA (Short Read Archive) of the National Center for Biotechnology Information (NCBI), under the submission number: SUB14602301." Please note that, though access restrictions are acceptable now, your entire minimal dataset will need to be made freely accessible if your manuscript is accepted for publication. This policy applies to all data except where public deposition would breach compliance with the protocol approved by your research ethics board.

**Reviewers' Comments:**

Reviewer's Responses to Questions

**Part I - Summary**

Reviewer #1: The manuscript has been much improved by the revisions. All my concerns have been addressed. I particularly appreciate the inclusion of pangenome graphs and analyses, providing more depth and better formalizing the findings in wheat powdery mildew.

Reviewer #2: The authors have thoroughly addressed all of my comments. The revised manuscript reflects substantial improvements in clarity, methodological detail, and interpretation. I appreciate the authors’ efforts in providing comprehensive responses and implementing the requested changes, which have strengthened the overall quality of the work.

Reviewer #3: This is a revised manuscript which I have previously reviewed. While the authors have made a significant number of changes and improved the manuscript, I feel they have chosen to not incorporate some major suggestions made independently by multiple reviewers. I disagree with some of their rational for not incorporating some changes and try to provide some detail on this below.

**Part II – Major Issues: Key Experiments Required for Acceptance**

Please use this section to detail the key new experiments or modifications of existing experiments that should be absolutely required to validate study conclusions.required to validate study conclusions.required to validate study conclusions.required to validate study conclusions.

Reviewer #1: (No Response)

Reviewer #2: (No Response)

Reviewer #3: I think the largest omission is still that the pangenome in terms of the proportion of core, softcore and fully accessory genes are still not reported, which the authors respond by saying the genome is too repetitive to achieve in an automated way with accuracy. While I agree with the authors that it is difficult to manually validate every gene presence absence polymorphism, I would also argue that having 100% accuracy in calling a gene present or absent in a pangenome is not expected (as the addition of new isolates could always highlight new “accessory” genes). I can only re-iterate my opinion that a pangenome study would attempt to quantify what proportion of genes are present in all isolates, a large fraction and genes that are singletons. I recognise that this comes with some degree of uncertainty. I believe the authors do have this data to hand (i.e. the genome annotations for each isolate), so I do not feel this is a major analyses to add in and would greatly benefit the study. The authors do present a pan-genome graph (as requested by another reviewer), showing areas of the genome that are more variable than others, primarily showing lots of heterogeneity around repetitive centromeric sequences. I guess I fail to see how these pangenome graphs really help drive our understanding of this pathogen, without some sort of more detailed analysis of the gene content within these loops.

Throughout the manuscript and especially towards the end of the results the grouping of candidate effectors on Chr11 is discussed. I found it difficult to understand if these are “candidate effectors” predicted based on some software or expression timing or if they were validated effectors. I noticed there was some inconsistency throughout the manuscript as to how these genes are referred to, in some cases “candidate” is used and in other cases not. The authors refer to a previous study that validated or classified these genes but I found this introduction quite late in the results. I wonder if this could be introduced earlier to clarify what this gene set is and whether or not these are still “candidate” or not. I also felt that these genes could be highlighted earlier in the manuscript which might help build towards the final results section looking more specifically at the presence absence of these genes. For example highlighting the (candidate) effectors locations in Fig2 (see below).

Reviewer Comment 3.12: Figure 2 remains unchanged

I do not feel that this comment has been addressed as Figure 2 remains unaltered. I think it is a relatively simple addition to add the centromere’s to Figure 2 that would enhance the manuscript and enable the text as written to match the Figure. While Figure 4 does show a lack of synteny around the centromere as indicated, why refer to this before the data is actually presented? Another enhancement of Figure2 I think would be to color the gene lines based on genes vs. candidate effectors. This would enable a global comparison of effector gene clustering on Chr11 which becomes much more important later in the manuscript.

Code used for R analyses and whole genome alignments is not provided. I think it is reasonable to ask for all code used to perform the population genetics and genome alignments to be provided as separate scripts in either the Zenodo repository or on GitHub. This is standard practice now within the field.

Linked to this comment the authors Methods lack details into how the synteny plots were produced, they refer to the plotting tool itself (NGenomeSym) but how the alignments between the genomes were generated remains slightly unclear. NGenomeSyn provides a few suggested methods for performing whole genome alignment including Mummer and Minimap2, but the authors do not state clearly what method was used. The code used is not provided so it is not possible for me to ascertain this information from the methods.

For the analysis of the sequence conservation accross isolates the authors provide a vague description of how they used blastn of 1kb segments to map synteny across the isolates. Again the code here is not provided and there are some important ommisions. For example blastn is quite sensitive so aside from looking at the “top blastn hit within 5% of its position in the reference chromosome”, were other thresholds used, for example query coverage and percent id? I would request that this custom blast script be provided for reproducibility.

Reviewer comment 2.7: Chr-Un

I agree with the other reviewers comment that artificially fusing contigs into a chimeric chromosome Un should not be done (i.e. has no biological meaning) and am surprised that the previous publications were allowed to do this (a small gap in peer review it seems). I think it would be worth pointing this out as politely as possible that this is problematic. Could the authors even use this manuscript to make the correction? Chr-Un still appears in many of the supplementary figures, which I would suggest removing throughout the manuscript.

The methods lines 948-950 in the new manuscript still contain a description of combining unplaced contigs, which I think should be removed. “Contigs with no blast hit were gathered into a chromosome Unknown” same again lines 987-898 “pieces of the mitochondrial genome were scattered in Bgt_chr-Un which we eventually did not use, while for the latter, the mitochondrial genome was in one contig”

Revised Discussion: While I appreciate that the authors have taken the reviewer comments on-board. I do not find some of these comments/ideas well integrated into the discussion and that the discussion still largely lacks a logical structure that is easy to follow. For example, in the first paragraph, they have added three examples of pan-genome studies and listed their general findings and simply follow this with a statement that pan-genomes from biotrophs are lacking. This is followed by a paragraph on host-specificity, followed by another paragraph that repeats some of the discussion on accessory chromosomes.

**Part III – Minor Issues: Editorial and Data Presentation Modifications**

Reviewer #1: (No Response)

Reviewer #2: (No Response)

Reviewer #3: New Comments:

Line 385-386: “In B.g. tritici isolates, chromosomes 1 through 10 show generally high sequence conservation”… Generally “high” sequence conservation is very generic, can this be more precisely presented with an average sequence identity using average ANI values presented in Table S7?

Line 722-72: Various used twice in same sentence, consider revising.

Methods:

FigS6: RII_Fuji is labelled RIX_Fuji in the main text (was corrected during revision) but remains RII in Supp materials.

FigS8: Why does Bgt_CHE_96224 have three versions for the BUSCO cds scores?

FigS9A: Green “ungrouped” effectors are not visible in this graph. Assuming numbers below indicate the number of genes? Perhaps an inset or a clearer legend could show these.

FigS11: There are still 12 Chromosomes in this figure. Assuming 12th is Chr-Un which should be removed.

FigS14A: Why do you have chloroplast (plant organelle) related genes in this figure legend?

PLOS authors have the option to publish the peer review history of their article (what does this mean?). If published, this will include your full peer review and any attached files.). If published, this will include your full peer review and any attached files.). If published, this will include your full peer review and any attached files.). If published, this will include your full peer review and any attached files.

...

Reviewer #1: **Yes:** Stefan KuschStefan KuschStefan KuschStefan Kusch

Reviewer #2: No

Reviewer #3: No

**Figure resubmission:**

**Reproducibility:**



---

## [Editor Report · Decision Letter 2]

2 Apr 2026

Dear Dr. Sotiropoulos,

We are pleased to inform you that your manuscript 'Combining pangenomics and population genetics finds chromosomal re-arrangements, diversified chromosome segments, copy number variations and transposon polymorphisms in wheat and rye powdery mildew' has been provisionally accepted for publication in PLOS Pathogens.

Best regards,

Michael F. Seidl

Academic Editor

PLOS Pathogens

Debra Bessen

Section Editor

PLOS Pathogens

Sumita Bhaduri-McIntosh

Editor-in-Chief

PLOS Pathogens

orcid.org/0000-0003-2946-9497

Michael Malim

Editor-in-Chief

PLOS Pathogens

orcid.org/0000-0002-7699-2064

The manuscript was reviewed mainly to address the remaining concerns of reviewer 3; reviewer 1 and 2 were fully satisfied in the previous round of revisions. I evaluated the changes to the manuscript and I conclude that these sufficiently address the reviewer's comments and suggestions. I have one last suggestion. To satisfy one of the reviewers comments, the manuscript now introduces the working definition of effectors very early. However, I feel that this is not very elegant and I would encourage the authors to consider to adjust this. One possibility could be to separate the general introduction in effectors/Avrs early in the manuscript with the working definition of effectors at a later and more appropriate moment. Ultimately, I leave this up to the authors, and I believe this could be addressed in the proof stage of the manuscript.
---

## [Editor Report · Acceptance letter]

Dear Dr. Sotiropoulos,

We are delighted to inform you that your manuscript, "Combining pangenomics and population genetics finds chromosomal re-arrangements, diversified chromosome segments, copy number variations and transposon polymorphisms in wheat and rye powdery mildew," has been formally accepted for publication in PLOS Pathogens.

Best regards,

Sumita Bhaduri-McIntosh

Editor-in-Chief

PLOS Pathogens

orcid.org/0000-0003-2946-9497

Michael Malim

Editor-in-Chief

PLOS Pathogens

orcid.org/0000-0002-7699-2064